# MetaOptimize: A Framework for Optimizing Step Sizes and Other Meta-parameters

**Arsalan Sharifnassab** [1] [2]   **Saber Salehkaleybar** [3]   **Richard Sutton** [2]

## Abstract

We address the challenge of optimizing meta-parameters (hyperparameters) in machine learning, a key factor for efficient training and high model performance. Rather than relying on expensive meta-parameter search methods, we introduce MetaOptimize: a dynamic approach that adjusts meta-parameters, particularly step sizes (also known as learning rates), during training. More specifically, MetaOptimize can wrap around any first-order optimization algorithm, tuning step sizes on the fly to minimize a specific form of regret that considers the long-term impact of step sizes on training, through a discounted sum of future losses. We also introduce lower-complexity variants of MetaOptimize that, in conjunction with its adaptability to various optimization algorithms, achieve performance comparable to those of the best hand-crafted learning rate schedules across diverse machine learning tasks.

## 1. Introduction

Optimization algorithms used in machine learning involve meta-parameters (i.e., hyperparameters) that substantially influence their performance. These meta-parameters are typically identified through a search process, such as grid search or other trial-and-error methods, prior to training. However, the computational cost of this meta-parameter search is significantly larger than that of training with optimal meta-parameters (Dahl et al., 2023; Jin, 2022). Meta-parameter optimization seeks to streamline this process by concurrently adjusting meta-parameters during training, moving away from the computationally expensive and often suboptimal trial and error search methods.

Meta-parameter optimization is particularly important in continual learning (De Lange et al., 2021), where continually changing environments or evolving loss functions necessitate adaptation of meta-parameters, such as step sizes, to track time-varying optima rather than settling on a static value.

In this work, we propose *MetaOptimize*, a general framework for optimizing meta-parameters to minimize a form of regret that explicitly accounts for the long-term influence of step sizes on future loss. Although this framework can handle various meta-parameters, we concentrate on step sizes as they are ubiquitous and crucial in practice.

MetaOptimize offers additional advantages beyond reducing search overhead. First, it enables dynamic step-size updates during training, potentially speeding the learning process. Traditional methods typically rely on manually designed learning rate schedules (e.g., initial increase followed by decay (Amid et al., 2022)), whereas MetaOptimize automatically discovers similar patterns.

Second, adapting step sizes across different network blocks (e.g., layers or neurons) can improve performance (Singh et al., 2015; Howard & Ruder, 2018), yet manually tuning such blockwise step sizes is impractical for large networks. By design, MetaOptimize handles these blockwise adjustments.

The concept of meta step-size optimization dates back to (Kesten, 1958), Delta-bar-Delta (Sutton, 1981; Jacobs, 1988), and its incremental variant, IDBD (Sutton, 1992). Numerous methods have emerged over the years (Section 8). This work distinguishes itself from prior efforts through the following key aspects:

- We introduce a formal approach to step-size optimization by minimizing a specific form of regret, essentially a discounted sum of future losses, and demonstrate how to do this causally via the MetaOptimize framework.

- MetaOptimize is general and can wrap around any first-order optimization algorithm (the *base update*), such as SGD, RMSProp (Hinton, 2012), Adam (Kingma & Ba,

[1]Openmind Research Institute, Canada [2]Department of Computing Science, University of Alberta, Edmonton, Canada [3]Leiden Institute of Advanced Computer Science, Leiden University, Leiden, Netherlands. Correspondence to: Arsalan Sharifnassab <arsalan.sharifnassab@openmindresearch.org>.

*Proceedings of the $42^{nd}$ International Conference on Machine Learning*, Vancouver, Canada. PMLR 267, 2025. Copyright 2025 by the author(s).

2014), or Lion (Chen et al., 2023), while optimizing step sizes via a separate first-order method (the *meta update*), such as SGD, Adam, RMSProp, or Lion.

- We develop approximation methods (Section 6) that, when incorporated into MetaOptimize, yield computationally efficient algorithms outperforming state-of-the-art automatic hyperparameter optimization methods on various stationary and continual (non-stationary) benchmarks (see Section 7).

- We show that some existing methods (like IDBD and its extensions, and hypergradient descent (Baydin et al., 2017)) are specific instances or approximations within the MetaOptimize framework (Section 5).

## 2. Problem Setting

We introduce a general continual optimization setting that, for a given sequence of loss functions $f_t(\cdot) : \mathbb{R}^n \to \mathbb{R}$, $t = 0, 1, 2, \ldots$, aims to find a sequence of weight vectors $\boldsymbol{w}_1, \boldsymbol{w}_2, \boldsymbol{w}_3, \ldots$ that minimize a discounted sum of future losses:

$$F_t^\gamma \stackrel{\text{def}}{=} (1 - \gamma) \sum_{\tau > t} \gamma^{\tau - t - 1} f_\tau(\boldsymbol{w}_\tau), \tag{1}$$

where $\gamma \in [0, 1)$ is a fixed discount factor, typically close to 1, called the *discount factor*. For stationary supervised learning, $f_t$ are i.i.d. samples from the same distribution, so minimizing $F_t^\gamma$ promotes rapid reduction of the expected loss.

Consider an arbitrary first-order optimization algorithm (e.g., SGD, RMSProp, Adam, or Lion) for updating $\boldsymbol{w}_t$. At time $t$, it takes the gradient $\nabla f_t(\boldsymbol{w}_t)$ of the current loss, along with an $m$-dimensional meta-parameter vector $\boldsymbol{\beta}_t$, to update $\boldsymbol{w}_t$ and possibly some internal variables $\tilde{\boldsymbol{x}}_t$ (e.g., momentum in Adam). Denoting $\boldsymbol{x}_t \stackrel{\text{def}}{=} \text{Stack}(\boldsymbol{w}_t, \tilde{\boldsymbol{x}}_t)$ and calling this update rule $\text{Alg}_{\text{base}}$, we have

$$\boldsymbol{x}_{t+1} = \text{Alg}_{\text{base}}(\boldsymbol{x}_t, \nabla f_t(\boldsymbol{w}_t), \boldsymbol{\beta}_t). \tag{2}$$

The goal of MetaOptimize is to determine a sequence $\boldsymbol{\beta}_t$ such that plugging them into the above base update yields a trajectory $\{\boldsymbol{w}_t\}$ minimizing $F_t^\gamma$.

Step-size adaptation is a natural special case: at each step $t$, the $m$-dimensional $\boldsymbol{\beta}_t$ defines an $n$-dimensional step-size vector $\boldsymbol{\alpha}_t$ via some fixed function $\sigma : \mathbb{R}^m \to \mathbb{R}^n$, i.e.,

$$\boldsymbol{\alpha}_t = \sigma(\boldsymbol{\beta}_t). \tag{3}$$

A good choice for $\sigma(\cdot)$ is exponential, ensuring $\boldsymbol{\alpha}_t$ is always positive and making multiplicative changes in $\boldsymbol{\alpha}_t$ correspond to additive changes in $\boldsymbol{\beta}_t$ (Sutton, 1992). By partitioning the network weights into blocks, we can learn a shared scalar step-size per block, or even a unique step-size per weight, all handled automatically by MetaOptimize.

## 3. Forward and Backward Views

Because $F_t^\gamma$ depends on future losses, minimizing it causally requires an alternative view. Suppose hypothetically we had oracle access to future information (i.e., future loss values and weights). We could update

$$\begin{aligned}
\boldsymbol{\beta}_{t+1} &= \boldsymbol{\beta}_t - \eta \frac{\mathrm{d}}{\mathrm{d}\boldsymbol{\beta}_t} F_t^\gamma \\
&= \boldsymbol{\beta}_t - \eta (1 - \gamma) \sum_{\tau > t} \gamma^{\tau - t - 1} \frac{\mathrm{d}}{\mathrm{d}\boldsymbol{\beta}_t} f_\tau(\boldsymbol{w}_\tau),
\end{aligned} \tag{4}$$

where $\eta$ is a meta step-size. This forward-view update, however, is not causal because we do not have the required future information at time $t$.

To address this, we adopt an eligibility-trace-style approach from reinforcement learning (Sutton, 1988; Sutton & Barto, 2018), introducing a *backward-view* update:

$$\boldsymbol{\beta}_{\tau+1} \leftarrow \boldsymbol{\beta}_\tau - \eta (1 - \gamma) \sum_{t < \tau} \gamma^{\tau - t - 1} \frac{\mathrm{d}}{\mathrm{d}\boldsymbol{\beta}_t} f_\tau(\boldsymbol{w}_\tau), \tag{5}$$

so that terms involving $f_\tau$ (and $\boldsymbol{w}_\tau$) appear at time $\tau$ (instead of $t$), which is the earliest time that these quantities become available. In the small-$\eta$ limit, the backward view closely approximates the forward view. [1]

Accordingly, we define a causal gradient estimate

$$\widehat{\nabla_{\boldsymbol{\beta}} F}_\tau \stackrel{\text{def}}{=} (1 - \gamma) \sum_{t=0}^{\tau-1} \gamma^{\tau - t - 1} \frac{\mathrm{d}}{\mathrm{d}\boldsymbol{\beta}_t} f_\tau(\boldsymbol{w}_\tau).$$

It follows from chain rule that

$$\widehat{\nabla_{\boldsymbol{\beta}} F}_\tau = \mathcal{H}_\tau^T \nabla f_\tau(\boldsymbol{w}_\tau), \tag{6}$$

where

$$\mathcal{H}_\tau \stackrel{\text{def}}{=} (1 - \gamma) \sum_{t=0}^{\tau-1} \gamma^{\tau - t - 1} \frac{d\boldsymbol{w}_\tau}{\mathrm{d}\boldsymbol{\beta}_t}. \tag{7}$$

Hence, $\mathcal{H}_\tau$ encodes how past $\boldsymbol{\beta}_t$ values cumulatively affect $\boldsymbol{w}_\tau$ under discounting by $\gamma$.

## 4. MetaOptimize

Algorithm 1 presents the general MetaOptimize framework for learning the meta-parameters $\boldsymbol{\beta}_t$. At each step

---

[1]Formally, assuming $f_t(\cdot) = 0$ for $t < 0$ and for $t > T$, the final updates under (5) and (4) become arbitrarily close as $\eta \to 0$. More specifically, $(\beta_T^{(5)} - \beta_T^{(4)})/\eta \to 0$, where $\beta_T^{(5)}$ and $\beta_T^{(4)}$ are the values of $\beta$ at time $T$ obtained from updates (5) and (4), respectively, starting from the same initial $\beta_0$. This is because as $\eta \to 0$, $\beta$ remains almost constant over $[0, T]$ interval, and the right hand sides of (5) and (4) match with accuracy $O(\eta^2)$, when summed over $[0, T]$. See Section 9 for a discussion on more accurate approximations for large $\eta$.

$t$, we replace the intractable gradient $\nabla_{\boldsymbol{\beta}} F_t^{\gamma}$ with the causal surrogate $\widehat{\nabla_{\boldsymbol{\beta}} F}_t$ to ensure the update is feasible in real time, as discussed in Section 3. Specifically, we feed $\widehat{\nabla_{\boldsymbol{\beta}} F}_t = \mathcal{H}_t^T \nabla f_t(\boldsymbol{w}_t)$ (from (6)) into any first-order meta-update rule $\text{Alg}_{\text{meta}}$, just like a conventional gradient.

Formally, define $\boldsymbol{y}_t \overset{\text{def}}{=} \text{Stack}(\boldsymbol{\beta}_t, \tilde{\boldsymbol{y}}_t)$ as the stack of meta-parameters $\boldsymbol{\beta}_t$ and any internal states $\tilde{\boldsymbol{y}}_t$ of $\text{Alg}_{\text{meta}}$ (e.g., momentum). The meta-update is then:

$$\boldsymbol{y}_{t+1} = \text{Alg}_{\text{meta}}\Big(\boldsymbol{y}_t, \mathcal{H}_t^T \nabla f_t(\boldsymbol{w}_t)\Big). \tag{8}$$

After applying the base update (2) to produce $\boldsymbol{x}_{t+1}$, we compute $\mathcal{H}_t^T \nabla f_t(\boldsymbol{w}_t)$ and plug it into (8) to update $\boldsymbol{y}_{t+1}$ (and thus $\boldsymbol{\beta}_{t+1}$). The remaining question is how to maintain $\mathcal{H}_t$ defined in (7), through application of the chain rule.

To compute $\mathcal{H}_t$ incrementally, let us stack the columns of the $n \times m$ matrix $\mathcal{H}_t$ into a single vector $\boldsymbol{h}_t$, and let

$$G_t \overset{\text{def}}{=} \begin{bmatrix} \frac{\text{d}\,\boldsymbol{y}_{t+1}}{\text{d}\,\boldsymbol{y}_t} & \frac{\text{d}\,\boldsymbol{y}_{t+1}}{\text{d}\,\boldsymbol{x}_t} & \frac{\text{d}\,\boldsymbol{y}_{t+1}}{\text{d}\,\boldsymbol{h}_t} \\ \frac{\text{d}\,\boldsymbol{x}_{t+1}}{\text{d}\,\boldsymbol{y}_t} & \frac{\text{d}\,\boldsymbol{x}_{t+1}}{\text{d}\,\boldsymbol{x}_t} & \frac{\text{d}\,\boldsymbol{x}_{t+1}}{\text{d}\,\boldsymbol{h}_t} \\ \frac{\text{d}\,\boldsymbol{h}_{t+1}}{\text{d}\,\boldsymbol{y}_t} & \frac{\text{d}\,\boldsymbol{h}_{t+1}}{\text{d}\,\boldsymbol{x}_t} & \frac{\text{d}\,\boldsymbol{h}_{t+1}}{\text{d}\,\boldsymbol{h}_t} \end{bmatrix}. \tag{9}$$

Applying the chain rule then implies

$$\begin{bmatrix} \frac{\text{d}\,\boldsymbol{y}_{t+1}}{\text{d}\,\boldsymbol{\beta}_\tau} \\ \frac{\text{d}\,\boldsymbol{x}_{t+1}}{\text{d}\,\boldsymbol{\beta}_\tau} \\ \frac{\text{d}\,\boldsymbol{h}_{t+1}}{\text{d}\,\boldsymbol{\beta}_\tau} \end{bmatrix} = G_t \begin{bmatrix} \frac{\text{d}\,\boldsymbol{y}_t}{\text{d}\,\boldsymbol{\beta}_\tau} \\ \frac{\text{d}\,\boldsymbol{x}_t}{\text{d}\,\boldsymbol{\beta}_\tau} \\ \frac{\text{d}\,\boldsymbol{h}_t}{\text{d}\,\boldsymbol{\beta}_\tau} \end{bmatrix},$$

which, when summed over $\tau$, turns into

$$\sum_{\tau=0}^{t} \gamma^{t-\tau} \begin{bmatrix} \frac{\text{d}\,\boldsymbol{y}_{t+1}}{\text{d}\,\boldsymbol{\beta}_\tau} \\ \frac{\text{d}\,\boldsymbol{x}_{t+1}}{\text{d}\,\boldsymbol{\beta}_\tau} \\ \frac{\text{d}\,\boldsymbol{h}_{t+1}}{\text{d}\,\boldsymbol{\beta}_\tau} \end{bmatrix} = G_t \begin{bmatrix} \frac{\text{d}\,\boldsymbol{y}_t}{\text{d}\,\boldsymbol{\beta}_t} \\ \frac{\text{d}\,\boldsymbol{x}_t}{\text{d}\,\boldsymbol{\beta}_t} \\ \frac{\text{d}\,\boldsymbol{h}_t}{\text{d}\,\boldsymbol{\beta}_t} \end{bmatrix} + G_t \sum_{\tau=0}^{t-1} \gamma^{t-\tau} \begin{bmatrix} \frac{\text{d}\,\boldsymbol{y}_t}{\text{d}\,\boldsymbol{\beta}_\tau} \\ \frac{\text{d}\,\boldsymbol{x}_t}{\text{d}\,\boldsymbol{\beta}_\tau} \\ \frac{\text{d}\,\boldsymbol{h}_t}{\text{d}\,\boldsymbol{\beta}_\tau} \end{bmatrix}. \tag{10}$$

Defining

$$Y_t \overset{\text{def}}{=} (1-\gamma) \sum_{\tau=0}^{t-1} \gamma^{t-\tau-1} \frac{\text{d}\,\boldsymbol{y}_t}{\text{d}\,\boldsymbol{\beta}_\tau} \tag{11}$$

$$X_t \overset{\text{def}}{=} (1-\gamma) \sum_{\tau=0}^{t-1} \gamma^{t-\tau-1} \frac{\text{d}\,\boldsymbol{x}_t}{\text{d}\,\boldsymbol{\beta}_\tau}, \tag{12}$$

$$Q_t \overset{\text{def}}{=} (1-\gamma) \sum_{\tau=0}^{t-1} \gamma^{t-\tau-1} \frac{\text{d}\,\boldsymbol{h}_t}{\text{d}\,\boldsymbol{\beta}_\tau}, \tag{13}$$

and noting that $\boldsymbol{y}_t = \text{Stack}(\boldsymbol{\beta}_t, \tilde{\boldsymbol{y}}_t)$, one obtains a compact update of the form

$$\begin{bmatrix} Y_{t+1} \\ X_{t+1} \\ Q_{t+1} \end{bmatrix} = G_t \left( \gamma \begin{bmatrix} Y_t \\ X_t \\ Q_t \end{bmatrix} + (1-\gamma) \begin{bmatrix} \begin{bmatrix} I \\ 0 \end{bmatrix} \\ 0 \\ 0 \end{bmatrix} \right), \tag{14}$$

---

**Algorithm 1** MetaOptimize Framework (for general meta-parameters)

**Given:** A base-update $\text{Alg}_{\text{base}}$, a meta-update $\text{Alg}_{\text{meta}}$, and a discount-factor $\gamma \le 1$.
**Initialize:**
$$X_0 = 0_{(n+\tilde{n}) \times m}, \quad Y_0 = \begin{bmatrix} I_{m \times m} \\ 0_{\tilde{m} \times m} \end{bmatrix}, \quad Q_0 = 0_{nm \times m}.$$
**for** $t = 0, 1, 2, \dots$ **do**
  $\boldsymbol{x}_{t+1} \leftarrow \text{Alg}_{\text{base}}(\boldsymbol{x}_t, \nabla f_t(\boldsymbol{w}_t), \boldsymbol{\beta}_t)$.
  $\mathcal{H}_t = $ first $n$ rows of $X_t$.
  $\boldsymbol{y}_{t+1} \leftarrow \text{Alg}_{\text{meta}}(\boldsymbol{y}_t, \mathcal{H}_t^T \nabla f_t(\boldsymbol{w}_t))$.
  Update $\begin{bmatrix} Y_{t+1} \\ X_{t+1} \\ Q_{t+1} \end{bmatrix}$ from (14), using $G_t$ in (9).
**end for**

---

and then extract $\mathcal{H}_t$ by taking the top $n$ rows of $X_t$ (since $\boldsymbol{x}_t = \text{Stack}(\boldsymbol{w}_t, \tilde{\boldsymbol{x}}_t)$). The blocks of $G_t$ can be found for standard algorithms (SGD, Adam, Lion, etc.) as detailed in Appendix A. Notably, the first row of $G_t$ blocks depends only on $\text{Alg}_{\text{meta}}$, and the rest of $G_t$ blocks depend only on $\text{Alg}_{\text{base}}$. Algorithm 1 summarizes the procedure.

*Remark* 4.1. A distinction of MetaOptimize from existing meta-parameter optimization methods is that it explicitly captures dynamics of the meta-parameters $\boldsymbol{\beta}$, and how changes in the current $\boldsymbol{\beta}$ affect $\boldsymbol{\beta}$ in future. The term $Y_t$ in (11) links changes in past $\boldsymbol{\beta}_t$ to future $\boldsymbol{\beta}$ values, which then influences $\mathcal{H}_t$. Intuitively, if $\boldsymbol{\beta}_t$ has been changing consistently in one direction (e.g., steadily increasing), it amplifies $Y_t$ and thus $\mathcal{H}_t$, accelerating ongoing updates. Conversely, if $\boldsymbol{\beta}_t$ stays nearly constant (indicating it may be close to optimal), $Y_t$ shrinks and so do the subsequent updates to $\boldsymbol{\beta}_t$, stabilizing around the optimum.

## 5. Reducing Complexity

The matrix $G_t$ can be large and may involve Hessian terms, increasing the computational burden. We discuss two practical approximations:

**2×2 approximation.** In (9), we zero out all blocks in the last row and column, effectively removing $Q_t$. Empirically, this simplification often has negligible impact on performance. Intuitively, $\mathcal{H}_t$ does not affect the base update directly ($\text{d}\,\boldsymbol{x}_{t+1}/\text{d}\,\boldsymbol{h}_t = 0$), so the extra blocks in $G_t$ involving $\boldsymbol{h}_t$ often have minor influence on the final meta-parameter trajectory.

**L-approximation.** We go one step further, also zeroing out the block in the first row and second column of $G_t$.

**Algorithm 2** MetaOptimize with $2 \times 2$ approx., $(\mathrm{Alg}_{\mathrm{base}}, \mathrm{Alg}_{\mathrm{meta}}) = (\mathrm{SGD}, \mathrm{SGD})$, and scalar step-size

---

**Initialize:** $\mathcal{H}_0 = \mathbf{0}_{n \times 1}, Y_0 = 1$.
**for** $t = 1, 2, \ldots$ **do**
    $\alpha_t = e^{\beta_t}$
    **Base update:**
      $\boldsymbol{w}_{t+1} = \boldsymbol{w}_t - \alpha_t \nabla f_t(\boldsymbol{w}_t)$
      $\mathcal{H}_{t+1} = \gamma\big(I - \alpha_t \nabla^2 f_t(\boldsymbol{w}_t)\big)\mathcal{H}_t - Y_t \alpha_t \nabla f_t(\boldsymbol{w}_t)$
      $Y_{t+1} = \gamma Y_t + (1 - \gamma) - \gamma \eta \mathcal{H}_t^T \nabla^2 f_t(\boldsymbol{w}_t) \mathcal{H}_t$
             `# For L-approximation let` $Y_{t+1} = 1$
    **Meta update:**
      $\beta_{t+1} = \beta_t - \eta \, \mathcal{H}_t^T \nabla f_t(\boldsymbol{w}_t)$
**end for**

---

Formally,

$$G_t^L \overset{\text{def}}{=} \begin{bmatrix} \frac{\mathrm{d}\,\boldsymbol{y}_{t+1}}{\mathrm{d}\,\boldsymbol{y}_t} & 0 \\ \frac{\mathrm{d}\,\boldsymbol{x}_{t+1}}{\mathrm{d}\,\boldsymbol{y}_t} & \frac{\mathrm{d}\,\boldsymbol{x}_{t+1}}{\mathrm{d}\,\boldsymbol{x}_t} \end{bmatrix}, \quad (15)$$

and the update in (14) simplifies to

$$\begin{bmatrix} Y_{t+1} \\ X_{t+1} \end{bmatrix} = G_t^L \left( \gamma \begin{bmatrix} Y_t \\ X_t \end{bmatrix} + (1 - \gamma) \begin{bmatrix} I \\ 0 \\ \hline 0 \end{bmatrix} \right). \quad (16)$$

This again discards $Q_t$, but also certain cross-terms in $Y_t$'s update. Empirically, L-approximation often matches the performance and sometimes improves the stability of the $2 \times 2$ approach.

**Intuition.** Algorithm 2 illustrates the $2 \times 2$ approximation for the case of SGD base/meta updates with a single scalar step size. Observe how $\mathcal{H}_t$ effectively accumulates (decayed) past gradients to decide whether to increase or decrease $\alpha_t$. If current and past gradients align, $\alpha_t$ is raised for faster learning; if they oppose each other, $\alpha_t$ shrinks. The decay $\gamma(I - [\boldsymbol{\alpha}] \nabla^2 f_t)$ of $\mathcal{H}_t$ ensures that if past gradients poorly approximate future ones due to large $\nabla^2 f_t$ or $\boldsymbol{\alpha}$, their influence fades more rapidly. Meanwhile, $Y_t$ reflects how changing past $\boldsymbol{\beta}$ influences the current $\boldsymbol{\beta}$; large swings in $\boldsymbol{\beta}$ amplify $\mathcal{H}_{t+1}$, while near-constant $\boldsymbol{\beta}$ dampens updates. Under the L-approximation, $Y_t$ becomes constant in this particular setup, further simplifying the algorithm.

**Containing some prior methods as special cases.** Under L-approximation, and restricting both base and meta updates to plain SGD, MetaOptimize reduces to IDBD (Sutton, 1982) and its extension (Xu et al., 2018); see Appendix B.1 for derivations. Another notable special case is $\gamma = 0$, which recovers the Hypergradient-descent approach (Baydin et al., 2017), updating step sizes to minimize the *immediate* loss $f_t(\boldsymbol{w}_t)$ rather than the discounted sum $F_t^\gamma$, ignoring long-term effects of step size on future loss.

## 6. Hessian-Free MetaOptimize

The $G_t$ matrix typically involves Hessian, $\nabla^2 f_t(\boldsymbol{w}_t)$, of the loss function, e.g., in the $d\boldsymbol{w}_{t+1}/d\boldsymbol{w}_t$ block where $\boldsymbol{w}_{t+1} = \boldsymbol{w}_t - \boldsymbol{\alpha}_t \nabla f_t(\boldsymbol{w}_t)$. Including second-order information in $G_t$ can be costly. Interestingly, for certain base and meta algorithms, we can eliminate the Hessian without much compromising the performance.

For example, Lion (Chen et al., 2023) updates weights by taking the sign of the gradient (plus momentum). Since the derivative of the sign function is zero almost everywhere, $\mathrm{d}\,\boldsymbol{w}_{t+1}/\mathrm{d}\,\boldsymbol{w}_t$ and related partials do not involve $\nabla^2 f_t(\boldsymbol{w}_t)$. Hence, *if both base and meta updates use Lion*, $G_t$ becomes Hessian-free throughout, avoiding second-order computations entirely (Appendix A.1.3, A.3.2).

For other algorithms, we may consider their *Hessian-free approximation* by zeroing out any Hessian term in $G_t$. The Hessian-free approximation turns out to be a good approximation, especially for base and meta algorithms that involve gradient normalization, like RMSProp and Adam. Note that, the sign function used in the Lion algorithm is an extreme form of normalization that divides a vector by its absolute value. We could instead use softer forms of normalization, such as normalizing to square root of a trace of squared vector, $\boldsymbol{v}_t$, as in RMSProp. Such normalizations typically result in two opposing Hessian-based terms in $\mathcal{H}_t$'s update (stemming from $\frac{\mathrm{d}\,\boldsymbol{w}_{t+1}}{\mathrm{d}\,\boldsymbol{w}_t}$ and $\frac{\mathrm{d}\,\boldsymbol{w}_{t+1}}{\mathrm{d}\,\boldsymbol{v}_t}$ blocks of matrix $G_t$), aiming to cancel out, particularly when consecutive gradients are positively correlated.

When Hessian terms are removed in the $2 \times 2$ approximation, $X_t$ and $Y_t$ become diagonal or simply vectorized, drastically reducing matrix-multiplication overhead. The overall complexity per step thus becomes similar to that of regular base and meta updates, requiring only a few extra vector operations. Algorithm 3 in Appendix A illustrates these Hessian-free variants (SGDm, AdamW, Lion) under $2 \times 2$ approximation.

In summary, Hessian-free and $2 \times 2$ or L-approximations yield a range of practical MetaOptimize instantiations that maintain strong performance at low additional cost.

## 7. Experiments

We evaluate MetaOptimize on image-classification and language-modeling benchmarks. Out of many possible base/meta-algorithm combinations and approximations (Algorithm 3), we showcase a few Hessian-free variants that performed well in practice. In the experiments, MetaOptimize starts with step-sizes set one or two orders of magnitude *below* typical good fixed step-sizes, with no specific tuning. We compare MetaOptimize against some popular baselines whose meta-parameters are well-tuned for each

task separately. See Appendix C for more details. Codes are available at https://github.com/sabersalehk/MetaOptimize.

## 7.1. CIFAR10 dataset

The first set of experiments involve training ResNet-18 with batch size of 100 on the CIFAR10 (Krizhevsky et al., 2009) dataset. Fig. 1 depicts the learning curves of four combinations of (base, meta) algorithms for Hessian-free MetaOptimize, along with the corresponding baselines with well-tuned fixed step sizes. Besides using a single scalar step-size, we also test a blockwise variant that partitions the ResNet18 parameters into six blocks (one for each linear layer and four blocks for the ResNet modules). In every tested combination, MetaOptimize outperforms its corresponding fixed-step-size baseline.

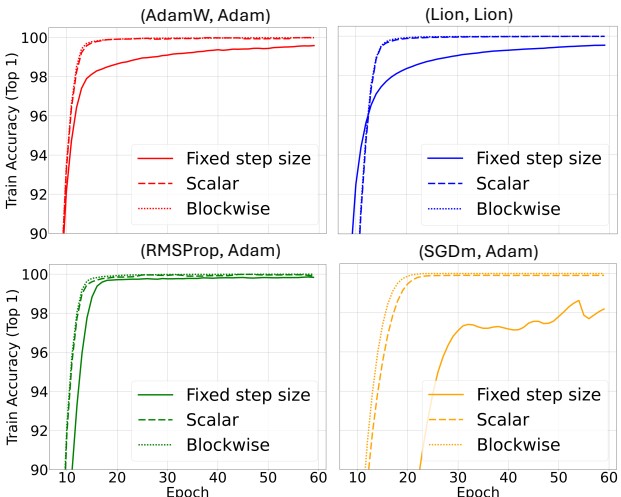

Figure 1. Learning curves for selected (base, meta) combinations in CIFAR10.

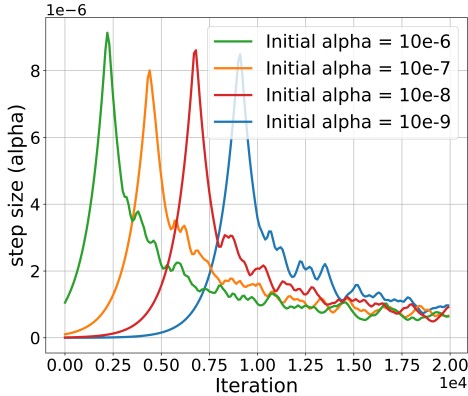

Figure 2. Robustness to initial step-sizes, for (Lion, Lion) as (base, meta) update in CIFAR10.

Interestingly, as demonstrated in Fig. 2, the MetaOptimize algorithms show remarkable robustness to initial step-size choices, even for initial step sizes that are several orders of magnitude smaller than the optimal fixed step-size.

## 7.2. Non-stationary CIFAR100

We evaluated MetaOptimize in a non-stationary setting with 10 sequential tasks, each containing 10 classes from CIFAR100. After training for one epoch on a task, it abruptly switches without explicit notification to the optimizer, and without resetting weights. We use a batch size of one, meaning each data point is seen exactly once. The model is based on a simple CNN network consisting of two convolution (and max pooling) layers followed by a fully connected layer. Each curve is averaged over 5 random seeds.

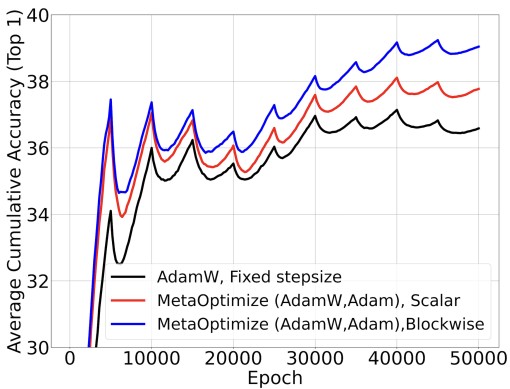

Figure 3. Learning curves for non-stationary CIFAR100.

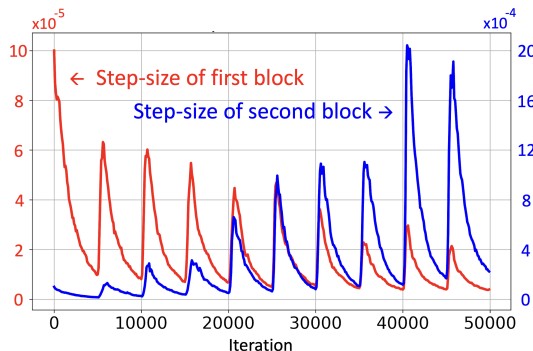

Figure 4. Blockwise stepsizes learned by MetaOptimize (AdamW,Adam) on non-stationary CIFAR100. Note that the scale of the y-axis for the two curves differ by an order of magnitude. Step-sizes of both blocks are initialized at $\alpha_0 = 10^{-4}$.

Figure 3 presents cumulative top-1 accuracy—averaged over all past training times—for AdamW (with the best fixed step-size), MetaOptimize with a scalar step-size, and MetaOptimize with blockwise step-sizes (two blocks: one for the first three layers and one for the last layer). MetaOptimize consistently outperforms the baseline. See Appendix D for additional plots.

The learned step-sizes reveal an interesting pattern (Fig. 4):

- **Task adaptation:** MetaOptimize increases step-sizes immediately after task switches to enhance adaptation.

- **Layer-wise behavior:** In blockwise case, early-layer step-sizes decrease over time, indicating convergence

---

**Algorithm 3** Hessian-free MetaOptimize algorithms with 2×2 approximation used in experiments

---

**Parameters:** $\eta > 0$ (default $10^{-3}$), $\gamma \in [0,1]$ (default $\simeq 1$)
**Initialize:** $\boldsymbol{h}_0 = \boldsymbol{0}_{n \times 1}$.
**for** $t = 1, 2, \ldots$ **do**

    **Base update**

    $\boldsymbol{\alpha}_t = \sigma(\boldsymbol{\beta}_t)$     `# exponential scalar/blockwise`
    $\boldsymbol{m}_{t+1} = \rho \boldsymbol{m}_t + (1 - \rho) \nabla f_t(\boldsymbol{w}_t)$
    **if** $\mathrm{Alg}_{\mathrm{base}}$ is SGDm **then**     $\Delta \boldsymbol{w} = -\boldsymbol{\alpha}_t \boldsymbol{m}_t - \kappa \boldsymbol{\alpha}_t \boldsymbol{w}_t$
    **if** $\mathrm{Alg}_{\mathrm{base}}$ is Lion **then**     $\Delta \boldsymbol{w} = -\boldsymbol{\alpha}_t \,\mathrm{Sign}\left(c\,\boldsymbol{m}_t + (1-c)\nabla f_t\right) - \kappa \boldsymbol{\alpha}_t \boldsymbol{w}_t$
    **if** $\mathrm{Alg}_{\mathrm{base}}$ is AdamW **then**     $\boldsymbol{v}_{t+1} = \lambda \boldsymbol{v}_t + (1 - \lambda) \nabla f_t(\boldsymbol{w}_t)^2$
                         $\Delta \boldsymbol{w} = -\boldsymbol{\alpha}_t \mu_t \boldsymbol{m}_t / \sqrt{\boldsymbol{v}_t} - \kappa \boldsymbol{\alpha}_t \boldsymbol{w}_t$     `# where` $\mu_t = \sqrt{1 - \lambda^t}/(1 - \rho^t)$

    $\boldsymbol{w}_{t+1} = \boldsymbol{w}_t + \Delta \boldsymbol{w}$
    $\boldsymbol{h}_{t+1} = \gamma(1 - \kappa \boldsymbol{\alpha}_t)\boldsymbol{h}_t + \Delta \boldsymbol{w}$

    **Meta update**

    $z = \boldsymbol{h}_t^\top \nabla f_t(\boldsymbol{w}_t)$     `# This is for scalar step-sizes.`
                           `# For blockwise, should compute sum of` $\boldsymbol{h}_t \nabla f_t(\boldsymbol{w}_t)$ `over each block.`
    $\bar{m}_{t+1} = \bar{\rho}\,\bar{m}_t + (1 - \bar{\rho})\,z$
    **if** $\mathrm{Alg}_{\mathrm{meta}}$ is Lion **then**     $\boldsymbol{\beta}_{t+1} = \boldsymbol{\beta}_t - \eta\,\mathrm{Sign}\left(\bar{c}\,\bar{m}_t + (1 - \bar{c})z\right)$
    **if** $\mathrm{Alg}_{\mathrm{meta}}$ is Adam **then**     $\bar{v}_{t+1} = \bar{\lambda}\,\bar{v}_t + (1 - \bar{\lambda})\,z^2$
                           $\boldsymbol{\beta}_{t+1} = \boldsymbol{\beta}_t - \eta\,\bar{\mu}_t \bar{m}_t / \sqrt{\bar{v}_t}$     `# where` $\bar{\mu}_t = \sqrt{1 - \bar{\lambda}^t}/(1 - \bar{\rho}^t)$

**end for**

---

to globally useful features, while last-layer step-sizes increase, reflecting the need to adapt to changing labels.

### 7.3. ImageNet dataset

We trained ResNet-18 with batch-size 256 on ImageNet (Deng et al., 2009). We compared MetaOptimize with scalar step-size against four state-of-the-art hyperparamter optimization algorithms, namely DoG (Ivgi et al., 2023), gdtuo (Chandra et al., 2022), Prodigy (Mishchenko & Defazio, 2023), and mechanic (Cutkosky et al., 2024), as well as AdamW and Lion baselines with fixed step-sizes, and AdamW with a well-tuned cosine decay learning rate scheduler with a 10k iterations warmup. Learning curves and complexity overheads are shown respectively in Fig. 5 and Table 1, showcasing the advantage of MetaOptimize algorithms (learning curve of DoG is not depicted due to its relatively poor performance). Unlike CIFAR10, here the blockwise versions of MetaOptimize showed no improvement over the scalar versions. Refer to Appendix D for further details.

### 7.4. Language modeling

For language model experiments, we used the TinyStories dataset (Eldan & Li, 2023), a synthetic collection of brief stories designed for children aged 3 to 4. This dataset proves effective for training and evaluating language models that are significantly smaller than the current state-of-the-art, and capable of crafting stories that are not only fluent and coherent but also diverse.

We used the implementation in (Karpathy, 2024) for training

15M parameter model with a batch size of 128 on the TinyStories dataset. Two combinations of Hessian-free MetaOptimize with scalar step sizes were tested against Lion and AdamW with well-tuned fixed step sizes, AdamW with a well-tuned cosine decay learning rate scheduler with 1k warmup iterations, and the four state-of-the-art step-size adaptation algorithms mentioned in the previous subsection. According to the learning curves, shown in Fig. 6, MetaOptimize outperforms all baselines (with an initial delay due to small initial step-sizes), except for the well-tuned learning rate scheduler within 30k iterations.

### 7.5. Sensitivity analysis

Here, we briefly discussion the sensitivity of MetaOptimize to its meta-meta-parameters.

For the meta-stepsize $\eta$ in MetaOptimize, there is generally no need for tuning, and the default value $\eta = 10^{-3}$ works universally well in stationary supervised learning. All experiments in this section used this default value with no sweeping required. The rationale for this choice is that when using Adam, Lion, or RMSProp for meta-updates, the absolute change in $\beta$ per iteration is approximately $\eta \times O(1) \simeq 10^{-3}$. Unless the current stepsize $\alpha$ is already near its optimal value, most $\beta$ updates will consistently move toward the optimal $\beta$. Within 1,000 steps, $\beta$ can change by $O(1)$, nearly doubling or halving $\alpha = \exp(\beta)$. Over 10,000 iterations, $\alpha$ can adjust to stepsizes that are $e^{10} > 20,000$ times larger or smaller, allowing $\eta \simeq 10^{-3}$ to efficiently track optimal stepsizes while minimizing unnecessary fluctuations in $\alpha$.

Regarding the discount factor $\gamma$, we used the $\gamma = 1$ in all stationary experiments and observed minimal sensitivity to

*Table 1.* Per-iteration wall-clock-time and GPU-memory overhead (compared to AdamW).

| Algorithm | ImageNet | | TinyStories | |
|---|---|---|---|---|
| | Time | Space | Time | Space |
| AdamW (fixed stepsize) | 0% | 0% | 0% | 0% |
| DoG (Ivgi et al., 2023) | +45% | 1.4% | +268% | 0% |
| gdtuo (Chandra et al., 2022) | +85% | 64% | +150% | 21% |
| mechanic (Cutkosky et al., 2024) | +42% | 88% | +9% | 0% |
| Prodigy (Mishchenko & Defazio, 2023) | +42% | 13% | +9% | 0% |
| MetaOptimize (AdamW, Lion) | +44% | 33% | +13% | 0% |

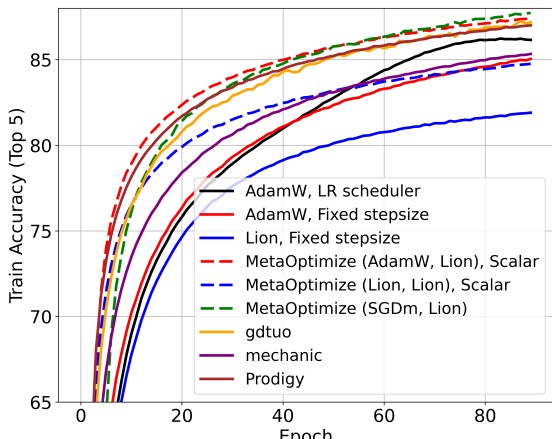

*Figure 5.* ImageNet learning curves.

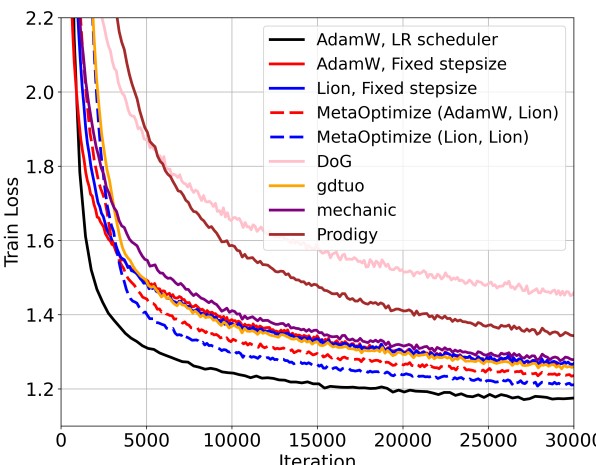

*Figure 6.* TinyStories (language model) learning curves.

$\gamma$ for values $\gamma \geq 0.999$ in a series of preliminary tests. However, performance begins to degrade with smaller values of $\gamma$. In the non-stationary CIFAR100, $\gamma = 0.999$ performed slightly better than 1.

## 8. Related Works

Automatic adaptation of step sizes, has been an important research topic in the literature of stochastic optimization. Several works aimed to remove the manual tuning of learning rates via adaptations of classical line search (Rolinek & Martius, 2018; Vaswani et al., 2019; Paquette & Scheinberg, 2020; Kunstner et al., 2023) and Polyak step size (Berrada et al., 2020; Loizou et al., 2021), stochastic proximal methods (Asi & Duchi, 2019), stochastic quadratic approximation (Schaul et al., 2013), hyper-gradient descent (Baydin et al., 2017), nested hyper-gradient descent (Chandra et al., 2022), distance to a solution adaptation (Ivgi et al., 2023; Defazio & Mishchenko, 2023; Mishchenko & Defazio, 2023), and online convex learning (Cutkosky et al., 2024). A limitation of most of these methods is their potential underperformance when their meta-parameters are not optimally configured for specific problems (Ivgi et al., 2023). Moreover, the primary focus of most of these methods is on minimizing immediate loss rather than considering the long-term effects of step sizes on future loss.

Normalization techniques proposed over past few years, such as AdaGrad (Duchi et al., 2011), RMSProp, and Adam have significantly enhanced the training process. While these algorithms show promise in the stationary problems, these normalization techniques do not optimize effective step sizes and are prone to have sub-optimal performance especially in the continual learning settings (Degris et al., 2024).

An early practical step-size optimization method was the Incremental-Delta-Bar-Delta (IDBD) algorithm, introduced in (Sutton, 1992), which aimed to optimize the step-size vector to minimize a specific form of quadratic loss functions in a continual setting. This algorithm was later extended for neural networks in (Xu et al., 2018; Donini et al., 2019), and further adapted in (Mahmood et al., 2012; Javed, 2020; Micaelli & Storkey, 2021) for different meta or base updates beyond SGD. However, the development of IDBD and its extensions included some implicit assumptions, notably overlooking the impact of step-size dynamics on the formulation of step-size update rules. These extensions are, in essence, special cases of the L-approximation within the MetaOptimize framework. The current work extends the IDBD research, significantly broadening the framework and establishing a solid basis for the derivations. IDBD and its extensions have been used in various machine learning tasks including independent component analysis (Schraudolph &

Giannakopoulos, 1999), human motion tracking (Kehl & Van Gool, 2006), classification (Koop, 2007), and reinforcement learning (Xu et al., 2018; Young et al., 2018; Javed et al., 2024). Refer to (Sutton, 2022) for a comprehensive history of step-size optimization.

Hypergradient Descent (HD) (Baydin et al., 2017) adapts learning rates using immediate loss gradients. MADA (Ozkara et al., 2024) extends HD by parameterizing a space of optimizers and navigating it via hypergradient descent. Both focus on short-term effects, whereas MetaOptimize introduces a discount factor $\gamma$ to model long-term influences, generalizing HD as a special case when $\gamma = 0$.

A related line of work is gradient-based bilevel optimization, initially introduced by (Bengio, 2000) and later expanded in (Maclaurin et al., 2015; Pedregosa, 2016; Franceschi et al., 2018; Gao et al., 2022). Recent advances, such as (Lorraine et al., 2020), enable the optimization of millions of hyperparameters. While bilevel optimization focuses on tuning hyperparameters to minimize validation loss through repeated full training runs of the base algorithm, MetaOptimize diverges significantly. Designed for continual learning, MetaOptimize optimizes meta-parameters on-the-fly during a single streaming run, without relying on validation loss. Instead, it minimizes online loss (or regret) directly, aligning with the continual learning framework where no validation or test sets exist, and data arrives sequentially.

Another relevant literature is learn to optimize (L2O), which aim to learn optimization strategies from data. Classical L2O methods such as (Andrychowicz et al., 2016) train optimizers offline and deploy them unchanged. Later works, including (Metz et al., 2020; 2022), develop more effective or scalable architectures, often using neural networks to modulate optimizer behavior. While powerful, these methods typically lack the ability to adapt online. In contrast, MetaOptimize updates its meta-parameters continuously during training, which is advantageous in nonstationary or continual learning scenarios.

There is also a line of research on the so-called parameter-free optimization that aims to remove the need for step-size tuning with almost no knowledge of the problem properties. Most of these methods are primarily designed for stochastic convex optimization (Luo & Schapire, 2015; Orabona & Pál, 2016), while more recent ones (Orabona & Tommasi, 2017; Ivgi et al., 2023) were applied to supervised learning tasks with small or moderate sample sizes.

## 9. Limitations and Future Works

Our work represents a step toward unlocking the potential of meta-parameter optimization, with substantial room for further exploration, some of which we outline here:

**Hessian:** We confined our experiments to Hessian-free methods for practicality, though Hessian-based algorithms could offer superior performance. These methods, however, face challenges requiring additional research. The Hessian matrix is notably noisy, impacting $\mathcal{H}_{t+1}$ multiplicatively, necessitating smoothing and clipping techniques. Additionally, the Hessian approximates the loss landscape's curvature but fails to account for non-differentiable curvatures, such as those from ReLU unit breakpoints, significant at training's end. From a computational perspective, developing low-complexity methods for approximate Hessian matrix products, especially for adjusting step-sizes at the layer and weight levels, is essential.

**More accurate traces:** As discussed in Section 3, accuracy of the backward approximation (5) may degrade for larger values of the meta-stepsize $\eta$. Eligibility traces in RL suffer from a similar problem, to resolve which more-sophisticated traces (e.g., Dutch traces) have been developed (see Chapter 11 of (Sutton & Barto, 2018)). Developing more accurate backward approximations for meta-parameter optimization can result in considerable improvements in performance and stability.

**Blockwise step-sizes:** While step sizes can vary much in granularity, our experiments focused on scalar and blockwise step-sizes. While increasing the number of step sizes is anticipated to enhance performance, our experimental findings in Section 7 reveal that this improvement is not consistent across the MetaOptimize approximations evaluated. Further investigation is needed in future research.

**Other approximations:** We explored a limited set of MetaOptimize's possible approximations, leaving a comprehensive analysis of various approximations for future research.

**Other meta-parameters:** Our study was limited to differentiable meta-parameters, not covering discrete ones like batch size or network layer count. We also did not investigate several significant differentiable meta-parameters beyond step-sizes, deferring such exploration to future work.

**Automatic Differentiation:** While certain versions of MetaOptimize, such as the L-Approximation, could be implemented using standard automatic differentiation software, its applicability to the general case of MetaOptimize remains unclear. Unlike updates for $w$ and $\beta$ (base and meta parameters), the $H$ matrix lacks an explicit incremental formula that can be easily handled by automatic differentiation. For some versions of MetaOptimize, including the Hessian-free approximations used in our experiments, automatic differentiation is unnecessary, as meta updates do not require additional differentiation. Exploring the scope and applicability of automatic differentiation across different MetaOptimize instances is an interesting direction for future research.

**Discount factor $\gamma = 1$:** Our backward formulation (Eq. (14)) formally assumes $\gamma < 1$ due to a normalization factor used in the definition of the surrogate gradient. For $\gamma = 1$, a simple workaround is to remove this scaling factor, which makes the derivation fully valid and consistent—matching our actual implementation and experiments. That said, the case $\gamma = 1$ presents subtle theoretical differences, much like in reinforcement learning and dynamic programming where additional centering is often helpful. Adapting similar techniques for meta-optimization may yield benefits and is a promising direction for future work.

## Impact Statement

This paper presents work whose goal is to advance the field of Machine Learning. There are many potential societal consequences of our work, none which we feel must be specifically highlighted here.

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

# Appendices

## A. Step-size Optimization for Different Choices of Base and Meta updates

In this appendix, we derive $G_t$ defined in (9) for different choices of algorithms for base and meta updates, and propose corresponding step-size optimization algorithms.

Consider the following partitions of $G_t$,

$$G_t^{\text{meta}} \stackrel{\text{def}}{=} \left[ \begin{array}{ccc} \dfrac{\mathrm{d}\,\boldsymbol{y}_{t+1}}{\mathrm{d}\,\boldsymbol{y}_t} & \dfrac{\mathrm{d}\,\boldsymbol{y}_{t+1}}{\mathrm{d}\,\boldsymbol{x}_t} & \dfrac{\mathrm{d}\,\boldsymbol{y}_{t+1}}{\mathrm{d}\,\boldsymbol{h}_t} \end{array} \right], \tag{17}$$

$$G_t^{\text{base}} \stackrel{\text{def}}{=} \left[ \begin{array}{ccc} \dfrac{\mathrm{d}\,\boldsymbol{x}_{t+1}}{\mathrm{d}\,\boldsymbol{y}_t} & \dfrac{\mathrm{d}\,\boldsymbol{x}_{t+1}}{\mathrm{d}\,\boldsymbol{x}_t} & \dfrac{\mathrm{d}\,\boldsymbol{x}_{t+1}}{\mathrm{d}\,\boldsymbol{h}_t} \\[2mm] \dfrac{\mathrm{d}\,\boldsymbol{h}_{t+1}}{\mathrm{d}\,\boldsymbol{y}_t} & \dfrac{\mathrm{d}\,\boldsymbol{h}_{t+1}}{\mathrm{d}\,\boldsymbol{x}_t} & \dfrac{\mathrm{d}\,\boldsymbol{h}_{t+1}}{\mathrm{d}\,\boldsymbol{h}_t} \end{array} \right]. \tag{18}$$

Then,

$$G_t = \left[ \begin{array}{ccc} \dfrac{\mathrm{d}\,\boldsymbol{y}_{t+1}}{\mathrm{d}\,\boldsymbol{y}_t} & \dfrac{\mathrm{d}\,\boldsymbol{y}_{t+1}}{\mathrm{d}\,\boldsymbol{x}_t} & \dfrac{\mathrm{d}\,\boldsymbol{y}_{t+1}}{\mathrm{d}\,\boldsymbol{h}_t} \\[2mm] \dfrac{\mathrm{d}\,\boldsymbol{x}_{t+1}}{\mathrm{d}\,\boldsymbol{y}_t} & \dfrac{\mathrm{d}\,\boldsymbol{x}_{t+1}}{\mathrm{d}\,\boldsymbol{x}_t} & \dfrac{\mathrm{d}\,\boldsymbol{x}_{t+1}}{\mathrm{d}\,\boldsymbol{h}_t} \\[2mm] \dfrac{\mathrm{d}\,\boldsymbol{h}_{t+1}}{\mathrm{d}\,\boldsymbol{y}_t} & \dfrac{\mathrm{d}\,\boldsymbol{h}_{t+1}}{\mathrm{d}\,\boldsymbol{x}_t} & \dfrac{\mathrm{d}\,\boldsymbol{h}_{t+1}}{\mathrm{d}\,\boldsymbol{h}_t} \end{array} \right] = \left[ \begin{array}{c} G_t^{\text{meta}} \\ \hline G_t^{\text{base}} \end{array} \right]. \tag{19}$$

In the sequel, we study base and meta updates separately, because $\text{Alg}_{\text{base}}$ and $\text{Alg}_{\text{meta}}$ impact disjoint sets of blocks in $G_t$. In particular, as we will see, the choice of $\text{Alg}_{\text{base}}$ only affects $G^{\text{base}}$ while the choice of $\text{Alg}_{\text{meta}}$ only affects $G^{\text{meta}}$.

**Notation conventions in all Appendices:** For any vector $\boldsymbol{v}$, we denote by $[\boldsymbol{v}]$ a diagonal matrix with diagonal entries derived from $\boldsymbol{v}$. We denote by $\sigma'(\boldsymbol{\beta}_t)$ the Jacobian of $\boldsymbol{\alpha}_t$ with respect to $\boldsymbol{\beta}_t$.

Before delving into computing $G_t^{\text{base}}$ and $G_t^{\text{meta}}$ for different base and meta algorithms, we further simplify these matrices.

### A.1. Derivation of $G^{\text{meta}}$ for Different Meta Updates

We start by simplifying $G^{\text{meta}}$, and introducing some notations.

Note that the meta update has no dependence on internal variables, $\tilde{\boldsymbol{x}}$, of the base algorithm. As a result,

$$\frac{\mathrm{d}\,\boldsymbol{y}_{t+1}}{\mathrm{d}\,\tilde{\boldsymbol{x}}_t} = 0. \tag{20}$$

Then,

$$G_t^{\text{meta}} = \left[ \begin{array}{ccc} \dfrac{\mathrm{d}\,\boldsymbol{y}_{t+1}}{\mathrm{d}\,\boldsymbol{y}_t} & \dfrac{\mathrm{d}\,\boldsymbol{y}_{t+1}}{\mathrm{d}\,\boldsymbol{x}_t} & \dfrac{\mathrm{d}\,\boldsymbol{y}_{t+1}}{\mathrm{d}\,\boldsymbol{h}_t} \end{array} \right] = \left[ \begin{array}{cccc} \dfrac{\mathrm{d}\,\boldsymbol{y}_{t+1}}{\mathrm{d}\,\boldsymbol{y}_t} & \dfrac{\mathrm{d}\,\boldsymbol{y}_{t+1}}{\mathrm{d}\,\boldsymbol{w}_t} & \dfrac{\mathrm{d}\,\boldsymbol{y}_{t+1}}{\mathrm{d}\,\tilde{\boldsymbol{x}}_t} & \dfrac{\mathrm{d}\,\boldsymbol{y}_{t+1}}{\mathrm{d}\,\boldsymbol{h}_t} \end{array} \right] = \left[ \begin{array}{cccc} \dfrac{\mathrm{d}\,\boldsymbol{y}_{t+1}}{\mathrm{d}\,\boldsymbol{y}_t} & \dfrac{\mathrm{d}\,\boldsymbol{y}_{t+1}}{\mathrm{d}\,\boldsymbol{w}_t} & 0 & \dfrac{\mathrm{d}\,\boldsymbol{y}_{t+1}}{\mathrm{d}\,\boldsymbol{h}_t} \end{array} \right], \tag{21}$$

where the third equality is due to (20). Let

$$L_t \stackrel{\text{def}}{=} \left[ \begin{array}{c|c|c|c} \nabla f_t(\boldsymbol{w}_t)^T & 0 & 0 & 0 \\ \hline 0 & \nabla f_t(\boldsymbol{w}_t)^T & 0 & 0 \\ \hline 0 & 0 & \ddots & 0 \\ \hline 0 & 0 & 0 & \nabla f_t(\boldsymbol{w}_t)^T \end{array} \right] \begin{array}{l} \leftarrow 1 \\ \leftarrow 2 \\ \vdots \\ \leftarrow m \end{array} \tag{22}$$

and recall that $\boldsymbol{h}_t$ is a vectorization of $\mathcal{H}_t$. Then,

$$\mathcal{H}_t \nabla f_t(\boldsymbol{w}_t) = L_t \boldsymbol{h}_t. \tag{23}$$

We now proceed to derivation of $G^{\text{meta}}$ for different choices of $\text{Alg}_{\text{meta}}$.

### A.1.1. **Meta SGD**

Here, we consider SGD for the meta update (8),

$$\boldsymbol{\beta}_{t+1} = \boldsymbol{\beta}_t - \eta \widehat{\nabla_{\boldsymbol{\beta}} F}_t = \boldsymbol{\beta}_t - \eta \, \mathcal{H}_t^T \nabla f_t(\boldsymbol{w}_t), \tag{24}$$

where $\eta$ is a scalar, called the *meta step size*. In this case, $\boldsymbol{y}_t = \boldsymbol{\beta}_t$. It then follows from (24) that

$$\frac{\mathrm{d}\boldsymbol{\beta}_{t+1}}{\mathrm{d}\boldsymbol{h}_t} = -\eta \frac{\mathrm{d}}{\mathrm{d}\boldsymbol{h}_t}\left(\mathcal{H}_t^T \nabla f_t(\boldsymbol{w}_t)\right) = -\eta \frac{\mathrm{d}}{\mathrm{d}\boldsymbol{h}_t}\left(L_t \boldsymbol{h}_t\right) = -\eta L_t, \tag{25}$$

where the second equality is due to (23). Consequently, from (21), we obtain

$$\begin{aligned}
G_t^{\text{meta}} &= \begin{bmatrix} \frac{\mathrm{d}\boldsymbol{y}_{t+1}}{\mathrm{d}\boldsymbol{y}_t} & \frac{\mathrm{d}\boldsymbol{y}_{t+1}}{\mathrm{d}\boldsymbol{w}_t} & 0 & \frac{\mathrm{d}\boldsymbol{y}_{t+1}}{\mathrm{d}\boldsymbol{h}_t} \end{bmatrix} \\
&= \begin{bmatrix} \frac{\mathrm{d}\boldsymbol{\beta}_{t+1}}{\mathrm{d}\boldsymbol{\beta}_t} & \frac{\mathrm{d}\boldsymbol{\beta}_{t+1}}{\mathrm{d}\boldsymbol{w}_t} & 0 & \frac{\mathrm{d}\boldsymbol{\beta}_{t+1}}{\mathrm{d}\boldsymbol{h}_t} \end{bmatrix} \\
&= \begin{bmatrix} I & -\eta \mathcal{H}_t^T \nabla^2 f_t(\boldsymbol{w}_t) & 0 & -\eta L_t \end{bmatrix},
\end{aligned} \tag{26}$$

where the last equality follows from (25) and simple differentiations of (24). Here, $\nabla^2 f_t(\boldsymbol{w}_t)$ denotes the Hessian of $f_t$ at $\boldsymbol{w}_t$.

### A.1.2. **Meta Adam**

The meta update based on the Adam algorithm is as follows,

$$\begin{aligned}
\bar{\boldsymbol{m}}_{t+1} &= \bar{\rho}\,\bar{\boldsymbol{m}}_t + \mathcal{H}_t^T \nabla f_t(\boldsymbol{w}_t), \\
\bar{\boldsymbol{v}}_{t+1} &= \bar{\lambda}\,\boldsymbol{v}_t + \left(\mathcal{H}_t^T \nabla f_t(\boldsymbol{w}_t)\right)^2, \\
\bar{\mu}_t &= \left(\frac{1-\bar{\rho}}{1-\bar{\rho}^t}\right) \Big/ \sqrt{\frac{1-\bar{\lambda}}{1-\bar{\lambda}^t}}, \\
\boldsymbol{\beta}_{t+1} &= \boldsymbol{\beta}_t - \eta\,\bar{\mu}_t \frac{\bar{\boldsymbol{m}}_t}{\sqrt{\bar{\boldsymbol{v}}_t}}
\end{aligned} \tag{27}$$

where $\bar{\boldsymbol{m}}_t$ is the momentum vector, $\bar{\boldsymbol{v}}_t$ is the trace of squared surrogate-meta-gradient. Since Adam algorithm needs to keep track of $\boldsymbol{\beta}_t$, $\bar{\boldsymbol{m}}_t$, and $\bar{\boldsymbol{v}}_t$, we have

$$\boldsymbol{y}_t = \begin{bmatrix} \boldsymbol{\beta}_t \\ \bar{\boldsymbol{m}}_t \\ \bar{\boldsymbol{v}}_t \end{bmatrix}. \tag{28}$$

Recall the following notation convention at the end of the Introduction section: for any $k \geq 1$, and any $k$-dimensional vector $\boldsymbol{v} = [v_1, \ldots, v_k]$, we denote the the corresponding diagonal matrix by $[\boldsymbol{v}]$:

$$[\boldsymbol{v}] \stackrel{\text{def}}{=} \begin{bmatrix} v_1 & \cdots & 0 \\ \vdots & \ddots & \vdots \\ 0 & \cdots & v_k \end{bmatrix}. \tag{29}$$

Consequently, from (21), we obtain

$$
G_t^{\text{meta}} = \left[ \begin{array}{c|cc|c} \frac{\mathrm{d}\,\boldsymbol{y}_{t+1}}{\mathrm{d}\,\boldsymbol{y}_t} & \frac{\mathrm{d}\,\boldsymbol{y}_{t+1}}{\mathrm{d}\,\boldsymbol{w}_t} & 0 & \frac{\mathrm{d}\,\boldsymbol{y}_{t+1}}{\mathrm{d}\,\boldsymbol{h}_t} \end{array} \right]
$$

$$
= \left[ \begin{array}{ccc|cc|c}
\frac{\mathrm{d}\,\boldsymbol{\beta}_{t+1}}{\mathrm{d}\,\boldsymbol{\beta}_t} & \frac{\mathrm{d}\,\boldsymbol{\beta}_{t+1}}{\mathrm{d}\,\bar{\boldsymbol{m}}_t} & \frac{\mathrm{d}\,\boldsymbol{\beta}_{t+1}}{\mathrm{d}\,\bar{\boldsymbol{v}}_t} & \frac{\mathrm{d}\,\boldsymbol{\beta}_{t+1}}{\mathrm{d}\,\boldsymbol{w}_t} & 0 & \frac{\mathrm{d}\,\boldsymbol{\beta}_{t+1}}{\mathrm{d}\,\boldsymbol{h}_t} \\[4pt]
\frac{\mathrm{d}\,\bar{\boldsymbol{m}}_{t+1}}{\mathrm{d}\,\boldsymbol{\beta}_t} & \frac{\mathrm{d}\,\bar{\boldsymbol{m}}_{t+1}}{\mathrm{d}\,\bar{\boldsymbol{m}}_t} & \frac{\mathrm{d}\,\bar{\boldsymbol{m}}_{t+1}}{\mathrm{d}\,\bar{\boldsymbol{v}}_t} & \frac{\mathrm{d}\,\bar{\boldsymbol{m}}_{t+1}}{\mathrm{d}\,\boldsymbol{w}_t} & 0 & \frac{\mathrm{d}\,\bar{\boldsymbol{m}}_{t+1}}{\mathrm{d}\,\boldsymbol{h}_t} \\[4pt]
\frac{\mathrm{d}\,\bar{v}t+1}{\mathrm{d}\,\boldsymbol{\beta}_t} & \frac{\mathrm{d}\,\bar{v}t+1}{\mathrm{d}\,\bar{\boldsymbol{m}}_t} & \frac{\mathrm{d}\,\bar{v}t+1}{\mathrm{d}\,\bar{\boldsymbol{v}}_t} & \frac{\mathrm{d}\,\bar{v}t+1}{\mathrm{d}\,\boldsymbol{w}_t} & 0 & \frac{\mathrm{d}\,\bar{v}t+1}{\mathrm{d}\,\boldsymbol{h}_t}
\end{array} \right] \tag{30}
$$

$$
= \left[ \begin{array}{ccc|cc|c}
I & -\eta\bar{\mu}_t\left[\frac{1}{\sqrt{\bar{\boldsymbol{v}}_t}}\right] & \frac{\eta\bar{\mu}_t}{2}\left[\frac{\bar{\boldsymbol{m}}_t}{\bar{\boldsymbol{v}}_t^{1.5}}\right] & 0 & 0 & 0 \\[4pt]
0 & \bar{\rho}I & 0 & \mathcal{H}_t^T\nabla^2 f_t & 0 & \frac{\mathrm{d}\,\bar{\boldsymbol{m}}_{t+1}}{\mathrm{d}\,\boldsymbol{h}_t} \\[4pt]
0 & 0 & \bar{\lambda}I & 2\left[\mathcal{H}_t^T\nabla f_t\right]\mathcal{H}_t^T\nabla^2 f_t & 0 & \frac{\mathrm{d}\,\bar{\boldsymbol{v}}_{t+1}}{\mathrm{d}\,\boldsymbol{h}_t}
\end{array} \right],
$$

where the last equality follows by calculating derivatives of (27). For the two remaining terms in the last column of $G_t$, we have

$$
\frac{\mathrm{d}\,\bar{\boldsymbol{m}}_{t+1}}{\mathrm{d}\,\boldsymbol{h}_t} = \frac{\mathrm{d}}{\mathrm{d}\,\boldsymbol{h}_t}\left(\mathcal{H}_t^T\nabla f_t(\boldsymbol{w}_t)\right) = \eta\frac{\mathrm{d}}{\mathrm{d}\,\boldsymbol{h}_t}\left(L_t\boldsymbol{h}_t\right) = \eta\,L_t. \tag{31}
$$

where the first equality follows from the update of $\bar{\boldsymbol{m}}_{t+1}$ in (27), and the second equality is due to (23). In the same vein,

$$
\frac{\mathrm{d}\,\bar{\boldsymbol{v}}_{t+1}}{\mathrm{d}\,\boldsymbol{h}_t} = \frac{\mathrm{d}}{\mathrm{d}\,\boldsymbol{h}_t}\left(\mathcal{H}_t^T\nabla f_t(\boldsymbol{w}_t)\right)^2 = \frac{\mathrm{d}}{\mathrm{d}\,\boldsymbol{h}_t}\left(L_t\boldsymbol{h}_t\right)^2 = 2\left[L_t\boldsymbol{h}_t\right]\frac{\mathrm{d}}{\mathrm{d}\,\boldsymbol{h}_t}\left(L_t\boldsymbol{h}_t\right) = 2\left[L_t\boldsymbol{h}_t\right]L_t = 2\left[\mathcal{H}_t^T\nabla f_t(\boldsymbol{w}_t)\right]L_t, \tag{32}
$$

where the first equality follows from the update of $\bar{\boldsymbol{v}}_{t+1}$ in (27), the second equality is due to (23), and the last equality is again from (23).

Plugging (31) and (32) into (30), we obtain

$$
G_t^{\text{meta}} = \left[ \begin{array}{ccc|cc|c}
I & -\eta\bar{\mu}_t\left[\frac{1}{\sqrt{\bar{\boldsymbol{v}}_t}}\right] & \frac{\eta\bar{\mu}_t}{2}\left[\frac{\bar{\boldsymbol{m}}_t}{\bar{\boldsymbol{v}}_t^{1.5}}\right] & 0 & 0 & 0 \\[4pt]
0 & \bar{\rho}I & 0 & \mathcal{H}_t^T\nabla^2 f_t & 0 & \eta\,L_t \\[4pt]
0 & 0 & \bar{\lambda}I & 2\left[\mathcal{H}_t^T\nabla f_t\right]\mathcal{H}_t^T\nabla^2 f_t & 0 & 2\left[\mathcal{H}_t^T\nabla f_t\right]L_t
\end{array} \right]. \tag{33}
$$

### A.1.3. **Meta Lion**

The meta update based on the lion algorithm is as follows

$$
\bar{\boldsymbol{m}}_{t+1} = \rho\,\bar{\boldsymbol{m}}_t + (1-\rho)\,\widehat{\nabla_{\boldsymbol{\beta}}F}_t, \tag{34}
$$

$$
\boldsymbol{\beta}_{t+1} = \boldsymbol{\beta}_t - \eta\,\text{Sign}\left(c\,\bar{\boldsymbol{m}}_t + (1-c)\widehat{\nabla_{\boldsymbol{\beta}}F}_t\right), \tag{35}
$$

where $\eta$ is a scalar, called the *meta step size*, and $\rho, c \in [0,1)$. Note that the meta algorithm operates on a low dimensional space. Therefore, we drop the regularizers like weight-decay in the meta updates, as they are primarily aimed to resolve the overfitting problem in high dimensional problems. Substituting $\widehat{\nabla_{\boldsymbol{\beta}}F}_t$ with $\mathcal{H}_t^T\nabla f_t(\boldsymbol{w}_t)$ we obtain the following meta updates

$$
\bar{\boldsymbol{m}}_{t+1} = \rho\,\bar{\boldsymbol{m}}_t + (1-\rho)\,\mathcal{H}_t^T\nabla f_t(\boldsymbol{w}_t), \tag{36}
$$

$$
\boldsymbol{\beta}_{t+1} = \boldsymbol{\beta}_t - \eta\,\text{Sign}\left(c\,\bar{\boldsymbol{m}}_t + (1-c)\mathcal{H}_t^T\nabla f_t(\boldsymbol{w}_t)\right). \tag{37}
$$

In this case,

$$
\boldsymbol{y}_t = \left[ \begin{array}{c} \boldsymbol{\beta}_t \\ \bar{\boldsymbol{m}}_t \end{array} \right],
$$

and

$$
\begin{aligned}
G_t^{\text{meta}} &= \left[\begin{array}{cccc} \frac{\mathrm{d}\, \boldsymbol{y}_{t+1}}{\mathrm{d}\, \boldsymbol{y}_t} & \frac{\mathrm{d}\, \boldsymbol{y}_{t+1}}{\mathrm{d}\, \boldsymbol{w}_t} & 0 & \frac{\mathrm{d}\, \boldsymbol{y}_{t+1}}{\mathrm{d}\, \boldsymbol{h}_t} \end{array}\right] \\[2mm]
&= \left[\begin{array}{ccccc} \frac{\mathrm{d}\, \boldsymbol{\beta}_{t+1}}{\mathrm{d}\, \boldsymbol{\beta}_t} & \frac{\mathrm{d}\, \boldsymbol{\beta}_{t+1}}{\mathrm{d}\, \bar{\boldsymbol{m}}_t} & \frac{\mathrm{d}\, \boldsymbol{\beta}_{t+1}}{\mathrm{d}\, \boldsymbol{w}_t} & 0 & \frac{\mathrm{d}\, \boldsymbol{\beta}_{t+1}}{\mathrm{d}\, \boldsymbol{h}_t} \\[2mm] \frac{\mathrm{d}\, \bar{\boldsymbol{m}}_{t+1}}{\mathrm{d}\, \boldsymbol{\beta}_t} & \frac{\mathrm{d}\, \bar{\boldsymbol{m}}_{t+1}}{\mathrm{d}\, \bar{\boldsymbol{m}}_t} & \frac{\mathrm{d}\, \bar{\boldsymbol{m}}_{t+1}}{\mathrm{d}\, \boldsymbol{w}_t} & 0 & \frac{\mathrm{d}\, \bar{\boldsymbol{m}}_{t+1}}{\mathrm{d}\, \boldsymbol{h}_t} \end{array}\right] \\[2mm]
&= \left[\begin{array}{ccccc} I & 0 & 0 & 0 & 0 \\[2mm] \frac{\mathrm{d}\, \bar{\boldsymbol{m}}_{t+1}}{\mathrm{d}\, \boldsymbol{\beta}_t} & \frac{\mathrm{d}\, \bar{\boldsymbol{m}}_{t+1}}{\mathrm{d}\, \bar{\boldsymbol{m}}_t} & \frac{\mathrm{d}\, \bar{\boldsymbol{m}}_{t+1}}{\mathrm{d}\, \boldsymbol{w}_t} & 0 & \frac{\mathrm{d}\, \bar{\boldsymbol{m}}_{t+1}}{\mathrm{d}\, \boldsymbol{h}_t} \end{array}\right],
\end{aligned}
\tag{38}
$$

where the last equality follows from (37). Consider the following block representation of $Y_t$:

$$
Y_t = \left[\begin{array}{c} B_t \\ Y_t^{\bar{m}} \end{array}\right].
\tag{39}
$$

Since the base algorithm, does not take $\bar{m}$ as input, as we will see in (41) and (42) of next subsection (Appendix A.2), $\frac{\mathrm{d}\, \bar{\boldsymbol{m}}_{t+1}}{\mathrm{d}\, \bar{\boldsymbol{m}}_t}$ is the only non-zero block of $G_t$ in its column of blocks (i.e., $\frac{\mathrm{d}\, s_{t+1}}{\mathrm{d}\, \bar{\boldsymbol{m}}_t} = 0$ for every variable $s$ other than $\bar{m}$). Consequently, it follows from (14) that $Y_t^{\bar{m}}$ as defined in (39), has no impact on the update of $X_{t+1}$, $B_{t+1}$, and $Q_{t+1}$. Therefore, we can zero-out the rows and columns of $G^{\text{meta}}$ that correspond to derivative of $\bar{m}$. As such we obtain the following equivalent of $G^{\text{meta}}$ in (38) from an algorithmic perspective:

$$
G_t^{\text{meta}} \equiv \left[\begin{array}{cc} I_{m \times m} & 0 \\ 0 & 0 \end{array}\right].
\tag{40}
$$

As a result, we get $B_t = I$ for all times $t$.

## A.2. Derivation of $G^{\text{base}}$ for Different Base Updates

We now turn our focus to computation of $G^{\text{base}}$ . Let us start by simplifying $G^{\text{base}}$, and introducing some notations.

Note that the base update has no dependence on internal variables, $\tilde{\boldsymbol{y}}$, of the meta update. As a result,

$$
\frac{\mathrm{d}\, \boldsymbol{x}_{t+1}}{\mathrm{d}\, \tilde{\boldsymbol{y}}_t} = 0.
\tag{41}
$$

Moreover, it follows from the definition of $\mathcal{H}_t$ in (7) that

$$
\frac{\mathrm{d}\, \mathcal{H}_{t+1}}{\mathrm{d}\, \tilde{\boldsymbol{y}}_t} = (1-\gamma) \sum_{t=0}^{t} \gamma^{t-\tau} \frac{\mathrm{d}}{d\tilde{\boldsymbol{y}}_t}\left(\frac{d\boldsymbol{w}_{t+1}}{\mathrm{d}\, \boldsymbol{\beta}_\tau}\right) = (1-\gamma) \sum_{t=0}^{t} \gamma^{t-\tau} \frac{\mathrm{d}}{d\boldsymbol{\beta}_\tau}\left(\frac{d\boldsymbol{w}_{t+1}}{\mathrm{d}\, \tilde{\boldsymbol{y}}_t}\right) = (1-\gamma) \sum_{t=0}^{t} \gamma^{t-\tau} \frac{\mathrm{d}}{d\boldsymbol{\beta}_\tau}\left(0\right) = 0,
$$

where the third equality follows from (41). Therefore,

$$
\frac{\mathrm{d}\, \boldsymbol{h}_{t+1}}{\mathrm{d}\, \tilde{\boldsymbol{y}}_t} = 0.
\tag{42}
$$

Note also that $\text{Alg}_{\text{base}}$ does not take $\mathcal{H}_t$ as input, and therefore,

$$
\frac{\mathrm{d}\, \boldsymbol{x}_{t+1}}{\mathrm{d}\, \boldsymbol{h}_t} = 0.
\tag{43}
$$

Consequently, we can simplify $G_t^{\text{base}}$ as follows,

$$
G_t^{\text{base}} = \left[\begin{array}{ccc} \frac{\mathrm{d}\, \boldsymbol{x}_{t+1}}{\mathrm{d}\, \boldsymbol{y}_t} & \frac{\mathrm{d}\, \boldsymbol{x}_{t+1}}{\mathrm{d}\, \boldsymbol{x}_t} & \frac{\mathrm{d}\, \boldsymbol{x}_{t+1}}{\mathrm{d}\, \boldsymbol{h}_t} \\[2mm] \frac{\mathrm{d}\, \boldsymbol{h}_{t+1}}{\mathrm{d}\, \boldsymbol{y}_t} & \frac{\mathrm{d}\, \boldsymbol{h}_{t+1}}{\mathrm{d}\, \boldsymbol{x}_t} & \frac{\mathrm{d}\, \boldsymbol{h}_{t+1}}{\mathrm{d}\, \boldsymbol{h}_t} \end{array}\right] = \left[\begin{array}{cccc} \frac{\mathrm{d}\, \boldsymbol{x}_{t+1}}{\mathrm{d}\, \boldsymbol{\beta}_t} & \frac{\mathrm{d}\, \boldsymbol{x}_{t+1}}{\mathrm{d}\, \tilde{\boldsymbol{y}}_t} & \frac{\mathrm{d}\, \boldsymbol{x}_{t+1}}{\mathrm{d}\, \boldsymbol{x}_t} & \frac{\mathrm{d}\, \boldsymbol{x}_{t+1}}{\mathrm{d}\, \boldsymbol{h}_t} \\[2mm] \frac{\mathrm{d}\, \boldsymbol{h}_{t+1}}{\mathrm{d}\, \boldsymbol{\beta}_t} & \frac{\mathrm{d}\, \boldsymbol{h}_{t+1}}{\mathrm{d}\, \tilde{\boldsymbol{y}}_t} & \frac{\mathrm{d}\, \boldsymbol{h}_{t+1}}{\mathrm{d}\, \boldsymbol{x}_t} & \frac{\mathrm{d}\, \boldsymbol{h}_{t+1}}{\mathrm{d}\, \boldsymbol{h}_t} \end{array}\right] = \left[\begin{array}{cccc} \frac{\mathrm{d}\, \boldsymbol{x}_{t+1}}{\mathrm{d}\, \boldsymbol{\beta}_t} & 0 & \frac{\mathrm{d}\, \boldsymbol{x}_{t+1}}{\mathrm{d}\, \boldsymbol{x}_t} & 0 \\[2mm] \frac{\mathrm{d}\, \boldsymbol{h}_{t+1}}{\mathrm{d}\, \boldsymbol{\beta}_t} & 0 & \frac{\mathrm{d}\, \boldsymbol{h}_{t+1}}{\mathrm{d}\, \boldsymbol{x}_t} & \frac{\mathrm{d}\, \boldsymbol{h}_{t+1}}{\mathrm{d}\, \boldsymbol{h}_t} \end{array}\right],
\tag{44}
$$

where the last equality is due to (41), (42), and (43).

On an independent note, consider the following block representation of $Y_t$,

$$Y_t = \left[ \begin{array}{c} B_t - \frac{1-\gamma}{\gamma} I \\ \tilde{Y}_t \end{array} \right], \tag{45}$$

Therefore,

$$\gamma Y_t + (1 - \gamma) \left[ \begin{array}{c} I \\ 0 \end{array} \right] = \gamma \left[ \begin{array}{c} B_t \\ \tilde{Y}_t \end{array} \right]$$

It then follows from (19) and (14) that

$$\left[ \begin{array}{c} X_{t+1} \\ Q_{t+1} \end{array} \right] = \gamma\, G_t^{\text{base}} \left[ \begin{array}{c} \left[ \begin{array}{c} B_t \\ \tilde{Y}_t \end{array} \right] \\ X_t \\ Q_t \end{array} \right]. \tag{46}$$

Moreover, from the definition of $Y_t$ in (11), we have

$$\frac{\mathrm{d}\, B_t}{\mathrm{d}\, \boldsymbol{x}_t} = (1 - \gamma) \frac{\mathrm{d}}{\mathrm{d}\, \boldsymbol{x}_t} \sum_{\tau=0}^{t} \gamma^{t-\tau} \frac{\mathrm{d}\, \boldsymbol{\beta}_t}{\mathrm{d}\, \boldsymbol{\beta}_\tau} = (1 - \gamma) \sum_{\tau=0}^{t} \gamma^{t-\tau} \frac{\mathrm{d}}{\mathrm{d}\, \boldsymbol{\beta}_\tau} \left( \frac{\mathrm{d}\, \boldsymbol{\beta}_t}{\mathrm{d}\, \boldsymbol{x}_t} \right) = (1 - \gamma) \sum_{\tau=0}^{t} \gamma^{t-\tau} \frac{\mathrm{d}}{\mathrm{d}\, \boldsymbol{\beta}_\tau} (0) = 0,$$

$$\frac{\mathrm{d}\, B_t}{\mathrm{d}\, \boldsymbol{\beta}_t} = (1 - \gamma) \frac{\mathrm{d}}{\mathrm{d}\, \boldsymbol{\beta}_t} \sum_{\tau=0}^{t} \gamma^{t-\tau} \frac{\mathrm{d}\, \boldsymbol{\beta}_t}{\mathrm{d}\, \boldsymbol{\beta}_\tau} = (1 - \gamma) \sum_{\tau=0}^{t} \gamma^{t-\tau} \frac{\mathrm{d}}{\mathrm{d}\, \boldsymbol{\beta}_\tau} \left( \frac{\mathrm{d}\, \boldsymbol{\beta}_t}{\mathrm{d}\, \boldsymbol{\beta}_t} \right) = (1 - \gamma) \sum_{\tau=0}^{t} \gamma^{t-\tau} \frac{\mathrm{d}}{\mathrm{d}\, \boldsymbol{\beta}_\tau} (I) = 0, \quad (47)$$

$$\frac{\mathrm{d}\, B_t}{\mathrm{d}\, \boldsymbol{h}_t} = (1 - \gamma) \frac{\mathrm{d}}{\mathrm{d}\, \boldsymbol{h}_t} \sum_{\tau=0}^{t} \gamma^{t-\tau} \frac{\mathrm{d}\, \boldsymbol{\beta}_t}{\mathrm{d}\, \boldsymbol{\beta}_\tau} = (1 - \gamma) \sum_{\tau=0}^{t} \gamma^{t-\tau} \frac{\mathrm{d}}{\mathrm{d}\, \boldsymbol{\beta}_\tau} \left( \frac{\mathrm{d}\, \boldsymbol{\beta}_t}{\mathrm{d}\, \boldsymbol{h}_t} \right) = (1 - \gamma) \sum_{\tau=0}^{t} \gamma^{t-\tau} \frac{\mathrm{d}}{\mathrm{d}\, \boldsymbol{\beta}_\tau} (0) = 0.$$

Finally, recall the definition

$$\sigma'(\boldsymbol{\beta}_t) \overset{\text{def}}{=} \frac{\mathrm{d}\, \boldsymbol{\alpha}_t}{\mathrm{d}\, \boldsymbol{\beta}_t} \tag{48}$$

as the Jacobian of $\boldsymbol{\alpha}_t$ with respect to $\boldsymbol{\beta}_t$.

We now proceed to derivation of $G^{\text{base}}$ for different choices of $\text{Alg}_{\text{base}}$.

### A.3. Base SGD

Base SGD algorithm makes the following base update in each iteration:

$$\boldsymbol{w}_{t+1} = \boldsymbol{w}_t - \boldsymbol{\alpha}_t \nabla f_t(\boldsymbol{w}_t). \tag{49}$$

In this case, $\boldsymbol{x}_t = \boldsymbol{w}_t$ and $X_t = \mathcal{H}_t$. Then, $G_t^{\text{base}}$ in (44) can be simplified to

$$
\begin{aligned}
G_t^{\text{base}} &= \left[ \begin{array}{cccc} \frac{\mathrm{d}\, \boldsymbol{x}_{t+1}}{\mathrm{d}\, \boldsymbol{\beta}_t} & 0 & \frac{\mathrm{d}\, \boldsymbol{x}_{t+1}}{\mathrm{d}\, \boldsymbol{x}_t} & 0 \\ \frac{\mathrm{d}\, \boldsymbol{h}_{t+1}}{\mathrm{d}\, \boldsymbol{\beta}_t} & 0 & \frac{\mathrm{d}\, \boldsymbol{h}_{t+1}}{\mathrm{d}\, \boldsymbol{x}_t} & \frac{\mathrm{d}\, \boldsymbol{h}_{t+1}}{\mathrm{d}\, \boldsymbol{h}_t} \end{array} \right] \\
&= \left[ \begin{array}{cccc} \frac{\mathrm{d}\, \boldsymbol{w}_{t+1}}{\mathrm{d}\, \boldsymbol{\beta}_t} & 0 & \frac{\mathrm{d}\, \boldsymbol{w}_{t+1}}{\mathrm{d}\, \boldsymbol{w}_t} & 0 \\ \frac{\mathrm{d}\, \boldsymbol{h}_{t+1}}{\mathrm{d}\, \boldsymbol{\beta}_t} & 0 & \frac{\mathrm{d}\, \boldsymbol{h}_{t+1}}{\mathrm{d}\, \boldsymbol{w}_t} & \frac{\mathrm{d}\, \boldsymbol{h}_{t+1}}{\mathrm{d}\, \boldsymbol{h}_t} \end{array} \right] \\
&= \left[ \begin{array}{cccc} -\left[\nabla f_t(\boldsymbol{w}_t)\right] \sigma'(\boldsymbol{\beta}_t) & 0 & I - [\boldsymbol{\alpha}_t] \nabla^2 f_t(\boldsymbol{w}_t) & 0 \\ \frac{\mathrm{d}\, \boldsymbol{h}_{t+1}}{\mathrm{d}\, \boldsymbol{\beta}_t} & 0 & \frac{\mathrm{d}\, \boldsymbol{h}_{t+1}}{\mathrm{d}\, \boldsymbol{w}_t} & \frac{\mathrm{d}\, \boldsymbol{h}_{t+1}}{\mathrm{d}\, \boldsymbol{h}_t} \end{array} \right],
\end{aligned} \tag{50}
$$

where the last equality follows by computing simple derivatives of $\boldsymbol{w}_{t+1}$ in (49).

We proceed to compute the three remaining entries of $G_t^{\text{base}}$, i.e., $\mathrm{d}\,\boldsymbol{h}_{t+1}/\mathrm{d}\,\boldsymbol{\beta}_t$, $\mathrm{d}\,\boldsymbol{h}_{t+1}/\mathrm{d}\,\boldsymbol{w}_t$, and $\mathrm{d}\,\boldsymbol{h}_{t+1}/\mathrm{d}\,\boldsymbol{h}_t$. Note that by plugging the first row of $G_t^{\text{base}}$, given in (50), into (46), and noting that $\mathcal{H}_t = X_t$, we obtain

$$\mathcal{H}_{t+1} = \gamma\big(I - [\boldsymbol{\alpha}_t]\,\nabla^2 f_t(\boldsymbol{w}_t)\big)\mathcal{H}_t \; - \; \gamma\,[\nabla f_t(\boldsymbol{w}_t)]\,\sigma'(\boldsymbol{\beta}_t)\,B_t, \tag{51}$$

for all $t \geq 0$. By vectorizing both sides of (51) we obtain

$$\boldsymbol{h}_{t+1} \;=\; \gamma\left[\begin{array}{c} \big(I - [\boldsymbol{\alpha}_t]\,\nabla^2 f_t\big)\,\mathcal{H}_t^{[1]} \;-\; [\nabla f_t]\,\sigma'(\boldsymbol{\beta}_t)\,B_t^{[1]} \\ \hline \big(I - [\boldsymbol{\alpha}_t]\,\nabla^2 f_t\big)\,\mathcal{H}_t^{[2]} \;-\; [\nabla f_t]\,\sigma'(\boldsymbol{\beta}_t)\,B_t^{[2]} \\ \hline \vdots \\ \hline \big(I - [\boldsymbol{\alpha}_t]\,\nabla^2 f_t\big)\,\mathcal{H}_t^{[m]} \;-\; [\nabla f_t]\,\sigma'(\boldsymbol{\beta}_t)\,B_t^{[m]} \end{array}\right]. \tag{52}$$

Note that for any pair of same-size vectors $\boldsymbol{a}$ and $\boldsymbol{b}$, we have $[\boldsymbol{a}]\,\boldsymbol{b} = [\boldsymbol{b}]\,\boldsymbol{a}$ where $[\boldsymbol{a}]$ and $[\boldsymbol{b}]$ are diagonal matrices of $\boldsymbol{a}$ and $\boldsymbol{b}$, respectively. Therefore, (52) can be equivalently written in the following form

$$\boldsymbol{h}_{t+1} \;=\; \gamma\left[\begin{array}{c} \big(I - [\boldsymbol{\alpha}_t]\,\nabla^2 f_t\big)\,\mathcal{H}_t^{[1]} \;-\; \big[\sigma'(\boldsymbol{\beta}_t)\,B_t^{[1]}\big]\,\nabla f_t \\ \hline \vdots \\ \hline \big(I - [\boldsymbol{\alpha}_t]\,\nabla^2 f_t\big)\,\mathcal{H}_t^{[m]} \;-\; \big[\sigma'(\boldsymbol{\beta}_t)\,B_t^{[m]}\big]\,\nabla f_t \end{array}\right]. \tag{53}$$

By taking the derivative of (52) with respect to $\boldsymbol{h}_t$, we obtain

$$\frac{\mathrm{d}\,\boldsymbol{h}_{t+1}}{\mathrm{d}\,\boldsymbol{h}_t} = \gamma\left[\begin{array}{c|c|c|c} I - [\boldsymbol{\alpha}_t]\,\nabla^2 f_t(\boldsymbol{w}_t) & 0 & 0 & 0 \\ \hline 0 & I - [\boldsymbol{\alpha}_t]\,\nabla^2 f_t(\boldsymbol{w}_t) & 0 & 0 \\ \hline 0 & 0 & \ddots & 0 \\ \hline 0 & 0 & 0 & I - [\boldsymbol{\alpha}_t]\,\nabla^2 f_t(\boldsymbol{w}_t) \end{array}\right] \begin{array}{l} \leftarrow \text{1st} \\[1.2em] \leftarrow \text{2nd} \\[1.2em] \vdots \\[1.2em] \leftarrow m\text{th} \end{array}. \tag{54}$$

In the above equation, note that $\mathrm{d}\,B_t/\mathrm{d}\,\boldsymbol{h}_t = 0$ due to (47). Let $\beta_t[i]$ and $w_t[j]$ denote the $i$th and $j$th entries of $\boldsymbol{\beta}_t$ and $\boldsymbol{w}_t$, for $i = 1, \ldots, m$ and $j = 1, \ldots, n$, respectively. It then follows from (52) and (47) that

$$\frac{\mathrm{d}\,\boldsymbol{h}_{t+1}}{\mathrm{d}\,\boldsymbol{\beta}_t} = -\gamma\left[\begin{array}{c|c|c} \big[\frac{\mathrm{d}\,\boldsymbol{\alpha}_t}{\mathrm{d}\,\beta_t[1]}\big]\nabla^2 f_t\,\mathcal{H}_t^{[1]} + [\nabla f_t]\frac{\partial\,\sigma'(\boldsymbol{\beta}_t)}{\partial\,\beta_t[1]}B_t^{[1]} & \cdots & \big[\frac{\mathrm{d}\,\boldsymbol{\alpha}_t}{\mathrm{d}\,\beta_t[m]}\big]\nabla^2 f_t\,\mathcal{H}_t^{[1]} + [\nabla f_t]\frac{\partial\,\sigma'(\boldsymbol{\beta}_t)}{\partial\,\beta_t[m]}B_t^{[1]} \\ \hline \vdots & \ddots & \vdots \\ \hline \big[\frac{\mathrm{d}\,\boldsymbol{\alpha}_t}{\mathrm{d}\,\beta_t[1]}\big]\nabla^2 f_t\,\mathcal{H}_t^{[m]} + [\nabla f_t]\frac{\partial\,\sigma'(\boldsymbol{\beta}_t)}{\partial\,\beta_t[1]}B_t^{[m]} & \cdots & \big[\frac{\mathrm{d}\,\boldsymbol{\alpha}_t}{\mathrm{d}\,\beta_t[m]}\big]\nabla^2 f_t\,\mathcal{H}_t^{[m]} + [\nabla f_t]\frac{\partial\,\sigma'(\boldsymbol{\beta}_t)}{\partial\,\beta_t[m]}B_t^{[m]} \end{array}\right], \tag{55}$$

where $\frac{\partial}{\partial\beta}$ stands for the entry-wise partial derivative of a matrix with respect to a scalar variable $\beta$. In the same vein, (53) and (47) imply that

$$\frac{\mathrm{d}\,\boldsymbol{h}_{t+1}}{\mathrm{d}\,\boldsymbol{w}_t} = -\gamma\left[\begin{array}{c} [\boldsymbol{\alpha}_t]\frac{\mathrm{d}\,(\nabla^2 f_t(\boldsymbol{w}_t)\,\mathcal{H}_t^{[1]})}{\mathrm{d}\,\boldsymbol{w}_t} + \big[\sigma'(\boldsymbol{\beta}_t)\,B_t^{[1]}\big]\nabla^2 f_t(\boldsymbol{w}_t) \\ \hline \vdots \\ \hline [\boldsymbol{\alpha}_t]\frac{\mathrm{d}\,(\nabla^2 f_t(\boldsymbol{w}_t)\,\mathcal{H}_t^{[m]})}{\mathrm{d}\,\boldsymbol{w}_t} + \big[\sigma'(\boldsymbol{\beta}_t)\,B_t^{[m]}\big]\nabla^2 f_t(\boldsymbol{w}_t) \end{array}\right]. \tag{56}$$

Finally, $G_t^{\text{base}}$ is obtained by plugging (54), (55), and (56) into (50).

In the **special case that $\beta$ is a scalar** (equivalently $m = 1$), and furthermore $\alpha = \sigma(\beta) = e^\beta$, matrix $G_t^{\text{base}}$ would be simplified to

$$G_t^{\text{base (scalar)}} = \begin{bmatrix} 1 & -\eta\, \boldsymbol{h}_t^T \nabla^2 f_t(\boldsymbol{w}_t) & -\eta\, \nabla f_t(\boldsymbol{w}_t)^T \\ -\alpha \nabla f_t(\boldsymbol{w}_t) & I - \alpha \nabla^2 f_t(\boldsymbol{w}_t) & 0 \\ -\gamma\alpha\nabla^2 f_t(\boldsymbol{w}_t)\boldsymbol{h}_t - B_t\,\alpha\nabla f_t(\boldsymbol{w}_t) & -\gamma\alpha\frac{\mathrm{d}\left(\nabla^2 f_t(\boldsymbol{w}_t)\boldsymbol{h}_t\right)}{d\boldsymbol{w}_t} - B_t\,\alpha\nabla^2 f_t(\boldsymbol{w}_t) & \gamma\left(I - \alpha\nabla^2 f_t(\boldsymbol{w}_t)\right) \end{bmatrix}.$$

### A.3.1. Base AdamW

The base update according to the AdamW algorithm (Loizou et al., 2021) is as follows,

$$\begin{aligned} \boldsymbol{m}_{t+1} &= \rho\, \boldsymbol{m}_t + \nabla f_t(\boldsymbol{w}_t), \\ \boldsymbol{v}_{t+1} &= \lambda\, \boldsymbol{v}_t + \nabla f_t(\boldsymbol{w}_t)^2, \\ \mu_t &= \left(\frac{1-\rho}{1-\rho^t}\right) \Big/ \sqrt{\frac{1-\lambda}{1-\lambda^t}}, \\ \boldsymbol{w}_{t+1} &= \boldsymbol{w}_t - \boldsymbol{\alpha}_t \mu_t \frac{\boldsymbol{m}_t}{\sqrt{\boldsymbol{v}_t}} - \kappa\boldsymbol{\alpha}_t\boldsymbol{w}_t, \end{aligned} \tag{57}$$

where $\boldsymbol{m}_t$ is the momentum vector, $\boldsymbol{v}_t$ is the trace of gradient square used for normalization, and $\kappa > 0$ is a weight-decay parameter. Therefore the base algorithm needs to keep track of $\boldsymbol{w}_t, \boldsymbol{m}_t, \boldsymbol{v}_t$, i.e.,

$$\boldsymbol{x}_t = \begin{bmatrix} \boldsymbol{w}_t \\ \boldsymbol{m}_t \\ \boldsymbol{v}_t \end{bmatrix}. \tag{58}$$

It then follows from (44) that

$$\begin{aligned} G_t^{\text{base}} &= \begin{bmatrix} \frac{\mathrm{d}\boldsymbol{x}_{t+1}}{\mathrm{d}\boldsymbol{\beta}_t} & 0 & \frac{\mathrm{d}\boldsymbol{x}_{t+1}}{\mathrm{d}\boldsymbol{x}_t} & 0 \\ \frac{\mathrm{d}\boldsymbol{h}_{t+1}}{\mathrm{d}\boldsymbol{\beta}_t} & 0 & \frac{\mathrm{d}\boldsymbol{h}_{t+1}}{\mathrm{d}\boldsymbol{x}_t} & \frac{\mathrm{d}\boldsymbol{h}_{t+1}}{\mathrm{d}\boldsymbol{h}_t} \end{bmatrix} \\[2mm] &= \left[\begin{array}{cc|ccc|c} \frac{\mathrm{d}\boldsymbol{w}_{t+1}}{\mathrm{d}\boldsymbol{\beta}_t} & 0 & \frac{\mathrm{d}\boldsymbol{w}_{t+1}}{\mathrm{d}\boldsymbol{w}_t} & \frac{\mathrm{d}\boldsymbol{w}_{t+1}}{\mathrm{d}\boldsymbol{m}_t} & \frac{\mathrm{d}\boldsymbol{w}_{t+1}}{\mathrm{d}\boldsymbol{v}_t} & 0 \\ \frac{\mathrm{d}\boldsymbol{m}_{t+1}}{\mathrm{d}\boldsymbol{\beta}_t} & 0 & \frac{\mathrm{d}\boldsymbol{m}_{t+1}}{\mathrm{d}\boldsymbol{w}_t} & \frac{\mathrm{d}\boldsymbol{m}_{t+1}}{\mathrm{d}\boldsymbol{m}_t} & \frac{\mathrm{d}\boldsymbol{m}_{t+1}}{\mathrm{d}\boldsymbol{v}_t} & 0 \\ \frac{\mathrm{d}\boldsymbol{v}_{t+1}}{\mathrm{d}\boldsymbol{\beta}_t} & 0 & \frac{\mathrm{d}\boldsymbol{v}_{t+1}}{\mathrm{d}\boldsymbol{w}_t} & \frac{\mathrm{d}\boldsymbol{v}_{t+1}}{\mathrm{d}\boldsymbol{m}_t} & \frac{\mathrm{d}\boldsymbol{v}_{t+1}}{\mathrm{d}\boldsymbol{v}_t} & 0 \\ \hline \frac{\mathrm{d}\boldsymbol{h}_{t+1}}{\mathrm{d}\boldsymbol{\beta}_t} & 0 & \frac{\mathrm{d}\boldsymbol{h}_{t+1}}{\mathrm{d}\boldsymbol{w}_t} & \frac{\mathrm{d}\boldsymbol{h}_{t+1}}{\mathrm{d}\boldsymbol{m}_t} & \frac{\mathrm{d}\boldsymbol{h}_{t+1}}{\mathrm{d}\boldsymbol{v}_t} & \frac{\mathrm{d}\boldsymbol{h}_{t+1}}{\mathrm{d}\boldsymbol{h}_t} \end{array}\right] \\[2mm] &= \left[\begin{array}{cc|ccc|c} -\mu_t\left[\frac{\boldsymbol{m}_t}{\sqrt{\boldsymbol{v}_t}} + \kappa\boldsymbol{w}_t\right]\sigma'(\boldsymbol{\beta}_t) & 0 & I - \kappa\left[\boldsymbol{\alpha}_t\right] & -\mu_t\left[\frac{\boldsymbol{\alpha}_t}{\sqrt{\boldsymbol{v}_t}}\right] & \frac{\mu_t}{2}\left[\frac{\boldsymbol{\alpha}_t\boldsymbol{m}_t}{\boldsymbol{v}_t^{1.5}}\right] & 0 \\ 0 & 0 & \nabla^2 f_t & \rho I & 0 & 0 \\ 0 & 0 & 2\left[\nabla f_t\right]\nabla^2 f_t & 0 & \lambda I & 0 \\ \hline \frac{\mathrm{d}\boldsymbol{h}_{t+1}}{\mathrm{d}\boldsymbol{\beta}_t} & 0 & \frac{\mathrm{d}\boldsymbol{h}_{t+1}}{\mathrm{d}\boldsymbol{w}_t} & \frac{\mathrm{d}\boldsymbol{h}_{t+1}}{\mathrm{d}\boldsymbol{m}_t} & \frac{\mathrm{d}\boldsymbol{h}_{t+1}}{\mathrm{d}\boldsymbol{v}_t} & \frac{\mathrm{d}\boldsymbol{h}_{t+1}}{\mathrm{d}\boldsymbol{h}_t} \end{array}\right] \end{aligned} \tag{59}$$

where the last equality follows from simple derivative computations in (57).

We proceed to compute the terms in the last row of the $G_t^{\text{base}}$ above. Consider the following block representation of $X_t$,

$$X_t = \begin{bmatrix} \mathcal{H}_t \\ X_t^m \\ X_t^v \end{bmatrix}, \tag{60}$$

Plugging the first row of $G_t^{\text{base}}$, given in (59), into (46), implies that

$$\mathcal{H}_{t+1} = -\gamma\mu_t\left[\frac{\boldsymbol{m}_t}{\sqrt{\boldsymbol{v}_t}} + \kappa\boldsymbol{w}_t\right]\sigma'(\boldsymbol{\beta}_t)B_t + \gamma\left(I - \kappa\left[\boldsymbol{\alpha}_t\right]\right)\mathcal{H}_t - \gamma\mu_t\left[\frac{\boldsymbol{\alpha}_t}{\sqrt{\boldsymbol{v}_t}}\right]X_t^m + \gamma\frac{\mu_t}{2}\left[\frac{\boldsymbol{\alpha}_t\boldsymbol{m}_t}{\boldsymbol{v}_t^{1.5}}\right]X_t^v. \tag{61}$$

for all $t \geq 0$. Note that for any pair of same-size vectors $\boldsymbol{a}$ and $\boldsymbol{b}$, we have $[\boldsymbol{a}]\boldsymbol{b} = [\boldsymbol{b}]\boldsymbol{a}$ where $[\boldsymbol{a}]$ and $[\boldsymbol{b}]$ are diagonal matrices of $\boldsymbol{a}$ and $\boldsymbol{b}$, respectively. Therefore, the $i$th column in the matrix equation (61) can be equivalently written as

$$\mathcal{H}_{t+1}^{[i]} = -\gamma\mu_t\Big[\sigma'(\boldsymbol{\beta}_t)B_t^{[i]}\Big]\frac{\boldsymbol{m}_t}{\sqrt{\boldsymbol{v}_t}} + \kappa\boldsymbol{w}_t + \gamma\big(I - \kappa[\boldsymbol{\alpha}_t]\big)\mathcal{H}_t^{[i]} - \gamma\mu_t\Big[X_t^{m\,[i]}\Big]\frac{\boldsymbol{\alpha}_t}{\sqrt{\boldsymbol{v}_t}} + \gamma\frac{\mu_t}{2}\Big[X_t^{v\,[i]}\Big]\frac{\boldsymbol{\alpha}_t\boldsymbol{m}_t}{\boldsymbol{v}_t^{1.5}}, \quad (62)$$

where $B_t^{[i]}$, $\mathcal{H}_t^{[i]}$, $X_t^{m\,[i]}$, and $X_t^{v\,[i]}$ stand for the $i$th columns of $B_t$, $\mathcal{H}_t$, $X_t^m$, and $X_t^v$, respectively. Following similar arguments as in (47), it is easy to show that

$$\begin{aligned}
\frac{\mathrm{d}\,X_t^m}{\mathrm{d}\,\boldsymbol{\beta}_t} &= \frac{\mathrm{d}\,X_t^v}{\mathrm{d}\,\boldsymbol{\beta}_t} = 0, \\
\frac{\mathrm{d}\,X_t^m}{\mathrm{d}\,\boldsymbol{w}_t} &= \frac{\mathrm{d}\,X_t^v}{\mathrm{d}\,\boldsymbol{w}_t} = 0, \\
\frac{\mathrm{d}\,X_t^m}{\mathrm{d}\,\boldsymbol{m}_t} &= \frac{\mathrm{d}\,X_t^v}{\mathrm{d}\,\boldsymbol{m}_t} = 0, \\
\frac{\mathrm{d}\,X_t^m}{\mathrm{d}\,\boldsymbol{v}_t} &= \frac{\mathrm{d}\,X_t^v}{\mathrm{d}\,\boldsymbol{v}_t} = 0, \\
\frac{\mathrm{d}\,X_t^m}{\mathrm{d}\,\boldsymbol{h}_t} &= \frac{\mathrm{d}\,X_t^v}{\mathrm{d}\,\boldsymbol{h}_t} = 0.
\end{aligned} \quad (63)$$

Note that $\boldsymbol{h}_t$ is an $nm$-dimensional vector derived from stacking the columns of $\mathcal{H}_t$. Therefore, we consider a block representation of $\boldsymbol{h}_t$ consisting of $m$ blocks, each of which corresponds to a column of $\mathcal{H}_t$. By taking the derivative of (61) with respect to $\boldsymbol{h}_t$, and using (63), we obtain

$$\frac{\mathrm{d}\,\boldsymbol{h}_{t+1}}{\mathrm{d}\,\boldsymbol{h}_t} = \gamma \begin{bmatrix} I - \kappa[\boldsymbol{\alpha}_t] & 0 & 0 & 0 \\ \hline 0 & I - \kappa[\boldsymbol{\alpha}_t] & 0 & 0 \\ \hline 0 & 0 & \ddots & 0 \\ \hline 0 & 0 & 0 & I - \kappa[\boldsymbol{\alpha}_t] \end{bmatrix} \begin{matrix} \leftarrow \text{1st} \\ \\ \leftarrow \text{2nd} \\ \\ \vdots \\ \\ \leftarrow m\text{th} \end{matrix}. \quad (64)$$

Let $\beta_t[i]$ and $w_t[j]$ denote the $i$th and $j$th entries of $\boldsymbol{\beta}_t$ and $\boldsymbol{w}_t$, for $i = 1, \ldots, m$ and $j = 1, \ldots, n$, respectively. Note that $\mathrm{d}\,\boldsymbol{h}_{t+1}/\mathrm{d}\,\boldsymbol{\beta}_t$ is a block matrix, in the form of an $m \times m$ array of $n \times 1$ blocks, $\frac{\mathrm{d}\,\boldsymbol{h}_{t+1}}{\mathrm{d}\,\boldsymbol{\beta}_t}[i,j] \stackrel{\text{def}}{=} \frac{\mathrm{d}\,\mathcal{H}_{t+1}^{[i]}}{\mathrm{d}\,\beta_t[j]}$, for $i, j = 1, \ldots, m$. It then follows from (61) and (63) that, for $i, j = 1, \ldots, m$,

$$\begin{aligned}
\frac{\mathrm{d}\,\boldsymbol{h}_{t+1}}{\mathrm{d}\,\boldsymbol{\beta}_t}[i,j] =\ & \frac{\mathrm{d}\,\mathcal{H}_{t+1}^{[i]}}{\mathrm{d}\,\beta_t[j]} \\
=\ & -\gamma\mu_t\Big[\frac{\boldsymbol{m}_t}{\sqrt{\boldsymbol{v}_t}} + \kappa\boldsymbol{w}_t\Big]\Big(\frac{\partial\,\sigma'(\boldsymbol{\beta}_t)}{\partial\,\beta_t[j]}\Big)B_t^{[i]} + \gamma\Big(I - \kappa\Big[\frac{\mathrm{d}\,\boldsymbol{\alpha}_t}{\mathrm{d}\,\beta_t[j]}\Big]\Big)\mathcal{H}_t^{[i]} \\
& -\gamma\mu_t\Big[\frac{1}{\sqrt{\boldsymbol{v}_t}}\Big]\Big[\frac{\mathrm{d}\,\boldsymbol{\alpha}_t}{\mathrm{d}\,\beta_t[j]}\Big]X_t^{m\,[i]} + \gamma\frac{\mu_t}{2}\Big[\frac{\boldsymbol{m}_t}{\boldsymbol{v}_t^{1.5}}\Big]\Big[\frac{\mathrm{d}\,\boldsymbol{\alpha}_t}{\mathrm{d}\,\beta_t[j]}\Big]X_t^{v\,[i]},
\end{aligned} \quad (65)$$

where $\frac{\partial}{\partial\beta}$ stands for the entry-wise partial derivative of a matrix with respect to a scalar variable $\beta$.

In the same vein, it follows from (62) and (63) that

$$\frac{\mathrm{d}\,\boldsymbol{h}_{t+1}}{\mathrm{d}\,\boldsymbol{w}_t} = -\gamma\mu_t\kappa \begin{bmatrix} \Big[\sigma'(\boldsymbol{\beta}_t)B_t^{[1]}\Big] \\ \hline \vdots \\ \hline \Big[\sigma'(\boldsymbol{\beta}_t)B_t^{[m]}\Big] \end{bmatrix}, \quad (66)$$

$$\frac{\mathrm{d}\,\boldsymbol{h}_{t+1}}{\mathrm{d}\,\boldsymbol{m}_t} = \gamma\mu_t \begin{bmatrix} \left[\frac{\boldsymbol{\alpha}_t X_t^{v\,[1]}}{2\,\boldsymbol{v}_t^{1.5}} - \frac{\sigma'(\boldsymbol{\beta}_t)B_t^{[1]}}{\sqrt{\boldsymbol{v}_t}}\right] \\ \vdots \\ \left[\frac{\boldsymbol{\alpha}_t X_t^{v\,[m]}}{2\,\boldsymbol{v}_t^{1.5}} - \frac{\sigma'(\boldsymbol{\beta}_t)B_t^{[m]}}{\sqrt{\boldsymbol{v}_t}}\right] \end{bmatrix}, \tag{67}$$

$$\frac{\mathrm{d}\,\boldsymbol{h}_{t+1}}{\mathrm{d}\,\boldsymbol{v}_t} = \frac{\gamma\mu_t}{2} \begin{bmatrix} \left[\frac{1}{\boldsymbol{v}_t^{1.5}}\right]\left[\left(\sigma'(\boldsymbol{\beta}_t)B_t^{[1]}\right)\boldsymbol{m}_t + \boldsymbol{\alpha}_t\,X_t^{m\,[1]} - \frac{3\boldsymbol{\alpha}_t\boldsymbol{m}_t X_t^{v\,[1]}}{2\,\boldsymbol{v}_t}\right] \\ \vdots \\ \left[\frac{1}{\boldsymbol{v}_t^{1.5}}\right]\left[\left(\sigma'(\boldsymbol{\beta}_t)B_t^{[m]}\right)\boldsymbol{m}_t + \boldsymbol{\alpha}_t\,X_t^{m\,[m]} - \frac{3\boldsymbol{\alpha}_t\boldsymbol{m}_t X_t^{v\,[m]}}{2\,\boldsymbol{v}_t}\right] \end{bmatrix}. \tag{68}$$

Finally, $G_t^{\text{base}}$ is obtained by plugging (64), (65), (66), (67), and (68) into (59).

### A.3.2. Base Lion

The lion algorithm, when used for base update, is as follows

$$\boldsymbol{m}_{t+1} = \rho\,\boldsymbol{m}_t + (1-\rho)\,\nabla f_t(\boldsymbol{w}_t), \tag{69}$$

$$\boldsymbol{w}_{t+1} = \boldsymbol{w}_t - \boldsymbol{\alpha}_t\,\mathrm{Sign}\left(c\,\boldsymbol{m}_t + (1-c)\nabla f_t\right) - \kappa\boldsymbol{\alpha}_t\boldsymbol{w}_t, \tag{70}$$

where $\boldsymbol{m}_t$ is called the momentum, $\kappa > 0$ is the weight-decay parameter, $\rho, c \in [0,1)$ are constants, and $\mathrm{Sign}(\cdot)$ is a function that computes entry-wise sign of a vector. Let

$$\boldsymbol{x}_t = \begin{bmatrix} \boldsymbol{w}_t \\ \boldsymbol{m}_t \end{bmatrix}. \tag{71}$$

It then follows from (44) that

$$G_t^{\text{base}} = \begin{bmatrix} \frac{\mathrm{d}\,\boldsymbol{x}_{t+1}}{\mathrm{d}\,\boldsymbol{\beta}_t} & 0 & \frac{\mathrm{d}\,\boldsymbol{x}_{t+1}}{\mathrm{d}\,\boldsymbol{x}_t} & 0 \\ \frac{\mathrm{d}\,\boldsymbol{h}_{t+1}}{\mathrm{d}\,\boldsymbol{\beta}_t} & 0 & \frac{\mathrm{d}\,\boldsymbol{h}_{t+1}}{\mathrm{d}\,\boldsymbol{x}_t} & \frac{\mathrm{d}\,\boldsymbol{h}_{t+1}}{\mathrm{d}\,\boldsymbol{h}_t} \end{bmatrix}$$

$$= \begin{bmatrix} \frac{\mathrm{d}\,\boldsymbol{w}_{t+1}}{\mathrm{d}\,\boldsymbol{\beta}_t} & 0 & \frac{\mathrm{d}\,\boldsymbol{w}_{t+1}}{\mathrm{d}\,\boldsymbol{w}_t} & \frac{\mathrm{d}\,\boldsymbol{w}_{t+1}}{\mathrm{d}\,\boldsymbol{m}_t} & 0 \\ \frac{\mathrm{d}\,\boldsymbol{m}_{t+1}}{\mathrm{d}\,\boldsymbol{\beta}_t} & 0 & \frac{\mathrm{d}\,\boldsymbol{m}_{t+1}}{\mathrm{d}\,\boldsymbol{w}_t} & \frac{\mathrm{d}\,\boldsymbol{m}_{t+1}}{\mathrm{d}\,\boldsymbol{m}_t} & 0 \\ \frac{\mathrm{d}\,\boldsymbol{h}_{t+1}}{\mathrm{d}\,\boldsymbol{\beta}_t} & 0 & \frac{\mathrm{d}\,\boldsymbol{h}_{t+1}}{\mathrm{d}\,\boldsymbol{w}_t} & \frac{\mathrm{d}\,\boldsymbol{h}_{t+1}}{\mathrm{d}\,\boldsymbol{m}_t} & \frac{\mathrm{d}\,\boldsymbol{h}_{t+1}}{\mathrm{d}\,\boldsymbol{h}_t} \end{bmatrix} \tag{72}$$

$$= \begin{bmatrix} -\left[\mathrm{Sign}\left(c\,\boldsymbol{m}_t + (1-c)\nabla f_t\right) + \kappa\boldsymbol{w}_t\right]\sigma'(\boldsymbol{\beta}_t) & 0 & I - \kappa\left[\boldsymbol{\alpha}_t\right] & 0 & 0 \\ \frac{\mathrm{d}\,\boldsymbol{m}_{t+1}}{\mathrm{d}\,\boldsymbol{\beta}_t} & & 0 & \frac{\mathrm{d}\,\boldsymbol{m}_{t+1}}{\mathrm{d}\,\boldsymbol{w}_t} & \frac{\mathrm{d}\,\boldsymbol{m}_{t+1}}{\mathrm{d}\,\boldsymbol{m}_t} & 0 \\ \frac{\mathrm{d}\,\boldsymbol{h}_{t+1}}{\mathrm{d}\,\boldsymbol{\beta}_t} & & 0 & \frac{\mathrm{d}\,\boldsymbol{h}_{t+1}}{\mathrm{d}\,\boldsymbol{w}_t} & \frac{\mathrm{d}\,\boldsymbol{h}_{t+1}}{\mathrm{d}\,\boldsymbol{m}_t} & \frac{\mathrm{d}\,\boldsymbol{h}_{t+1}}{\mathrm{d}\,\boldsymbol{h}_t} \end{bmatrix}$$

where the second equality is due to (71) and the last equality follows from (70). Consider the following block representation of $X_t$,

$$X_t = \begin{bmatrix} \mathcal{H}_t \\ X_t^m \end{bmatrix}. \tag{73}$$

Plugging the first row of $G_t^{\text{base}}$, given in (72), into (46), implies that

$$\mathcal{H}_{t+1} = -\gamma\left[\mathrm{Sign}\left(c\,\boldsymbol{m}_t + (1-c)\nabla f_t\right) + \kappa\boldsymbol{w}_t\right]\sigma'(\boldsymbol{\beta}_t)\,B_t + \gamma\left(I - \kappa\left[\boldsymbol{\alpha}_t\right]\right)\mathcal{H}_t \tag{74}$$

For simplicity of notation, we define the diagonal matrix $S_t$ as

$$S_t \stackrel{\text{def}}{=} \left[\mathrm{Sign}\left(c\,\boldsymbol{m}_t + (1-c)\nabla f_t\right) + \kappa\boldsymbol{w}_t\right]. \tag{75}$$

Then,

$$\boldsymbol{h}_{t+1} = \gamma \begin{bmatrix} -S_t\,\sigma'(\boldsymbol{\beta}_t)\,B_t^{[1]} \,+\, \gamma\big(I - \kappa\,[\boldsymbol{\alpha}_t]\big)\mathcal{H}_t^{[1]} \\ \vdots \\ -S_t\,\sigma'(\boldsymbol{\beta}_t)\,B_t^{[m]} \,+\, \gamma\big(I - \kappa\,[\boldsymbol{\alpha}_t]\big)\mathcal{H}_t^{[m]} \end{bmatrix} \tag{76}$$

It follows that

$$\frac{\mathrm{d}\,\boldsymbol{h}_{t+1}}{\mathrm{d}\,\boldsymbol{m}_t} = 0, \tag{77}$$

and

$$\frac{\mathrm{d}\,\boldsymbol{h}_{t+1}}{\mathrm{d}\,\boldsymbol{w}_t} = -\gamma \begin{bmatrix} [\boldsymbol{e}_1]\,\sigma'(\boldsymbol{\beta}_t)\,B_t^{[1]} & \cdots & [\boldsymbol{e}_n]\,\sigma'(\boldsymbol{\beta}_t)\,B_t^{[1]} \\ \vdots & \ddots & \vdots \\ [\boldsymbol{e}_1]\,\sigma'(\boldsymbol{\beta}_t)\,B_t^{[m]} & \cdots & [\boldsymbol{e}_n]\,\sigma'(\boldsymbol{\beta}_t)\,B_t^{[m]} \end{bmatrix}, \tag{78}$$

where $\boldsymbol{e}_i$ is the $i$th unit vector (i.e., an $n$-dimensional vector whose $i$th entry is 1 and all other entries are zero). Let $\beta_t[i]$ and $\mathcal{H}_t^{[i]}$ be the $i$th entry of $\boldsymbol{\beta}_t$ and $i$th column of $\mathcal{H}_t$, respectively, for $i = 1, \ldots, m$. Then,

$$\frac{\mathrm{d}\,\boldsymbol{h}_{t+1}}{\mathrm{d}\,\boldsymbol{\beta}_t} = -\gamma \begin{bmatrix} \gamma\kappa\Big[\frac{\mathrm{d}\,\boldsymbol{\alpha}_t}{\mathrm{d}\,\beta_t[1]}\Big]\mathcal{H}_t^{[1]} + S_t\frac{\partial\,\sigma'(\boldsymbol{\beta}_t)}{\partial\,\beta_t[1]}B_t^{[1]} & \cdots & \gamma\kappa\Big[\frac{\mathrm{d}\,\boldsymbol{\alpha}_t}{\mathrm{d}\,\beta_t[m]}\Big]\mathcal{H}_t^{[1]} + S_t\frac{\partial\,\sigma'(\boldsymbol{\beta}_t)}{\partial\,\beta_t[m]}B_t^{[1]} \\ \vdots & \ddots & \vdots \\ \gamma\kappa\Big[\frac{\mathrm{d}\,\boldsymbol{\alpha}_t}{\mathrm{d}\,\beta_t[1]}\Big]\mathcal{H}_t^{[m]} + S_t\frac{\partial\,\sigma'(\boldsymbol{\beta}_t)}{\partial\,\beta_t[1]}B_t^{[m]} & \cdots & \gamma\kappa\Big[\frac{\mathrm{d}\,\boldsymbol{\alpha}_t}{\mathrm{d}\,\beta_t[m]}\Big]\mathcal{H}_t^{[m]} + S_t\frac{\partial\,\sigma'(\boldsymbol{\beta}_t)}{\partial\,\beta_t[m]}B_t^{[m]} \end{bmatrix}, \tag{79}$$

and

$$\frac{\mathrm{d}\,\boldsymbol{h}_{t+1}}{\mathrm{d}\,\boldsymbol{h}_t} = \gamma \begin{bmatrix} I - \kappa\,[\boldsymbol{\alpha}_t] & 0 & 0 & 0 \\ 0 & I - \kappa\,[\boldsymbol{\alpha}_t] & 0 & 0 \\ 0 & 0 & \ddots & 0 \\ 0 & 0 & 0 & I - \kappa\,[\boldsymbol{\alpha}_t] \end{bmatrix} \begin{matrix} \leftarrow 1\text{st} \\ \leftarrow 2\text{nd} \\ \vdots \\ \leftarrow m\text{th} \end{matrix}. \tag{80}$$

It follows from (21), (72), and (77) that in the $G_t$ matrix, $\frac{\mathrm{d}\,\boldsymbol{m}_{t+1}}{\mathrm{d}\,\boldsymbol{m}_t}$ is the only non-zero block in its corresponding column of blocks. Consequently, it follows from (14) that $X_t^m$, as defined in (73), has no impact on the update of $\mathcal{H}_{t+1}$, $Y_{t+1}$, and $Q_{t+1}$. Therefore, the rows and columns of $G^{\text{base}}$ that correspond to derivative of $\boldsymbol{m}$ can be completely removed from $G^{\text{base}}$. By removing these rows and columns from $G^t$, the matrix update (14) simplifies to

$$\begin{bmatrix} Y_{t+1} \\ \mathcal{H}_{t+1} \\ Q_{t+1} \end{bmatrix} = \gamma \begin{bmatrix} \begin{bmatrix} \frac{\mathrm{d}\,\boldsymbol{y}_{t+1}}{\mathrm{d}\,\boldsymbol{y}_t} \\ -\big[\mathrm{Sign}\,(c\,\boldsymbol{m}_t + (1-c)\nabla f_t)\big]\sigma'(\boldsymbol{\beta}_t) \quad 0 \\ \frac{\mathrm{d}\,\boldsymbol{h}_{t+1}}{\mathrm{d}\,\boldsymbol{\beta}_t} \qquad\qquad 0 \end{bmatrix} \quad \begin{bmatrix} \frac{\mathrm{d}\,\boldsymbol{y}_{t+1}}{\mathrm{d}\,\boldsymbol{w}_t} & \frac{\mathrm{d}\,\boldsymbol{y}_{t+1}}{\mathrm{d}\,\boldsymbol{h}_t} \\ I - \kappa\,[\boldsymbol{\alpha}_t] & 0 \\ \frac{\mathrm{d}\,\boldsymbol{h}_{t+1}}{\mathrm{d}\,\boldsymbol{w}_t} & \frac{\mathrm{d}\,\boldsymbol{h}_{t+1}}{\mathrm{d}\,\boldsymbol{h}_t} \end{bmatrix} \end{bmatrix} \left( \begin{bmatrix} Y_t \\ \mathcal{H}_t \\ Q_t \end{bmatrix} + (1-\gamma)\begin{bmatrix} I \\ 0 \\ 0 \\ 0 \end{bmatrix} \right), \tag{81}$$

where $\mathrm{d}\,\boldsymbol{h}_{t+1}/\mathrm{d}\,\boldsymbol{\beta}_t$, $\mathrm{d}\,\boldsymbol{h}_{t+1}/\mathrm{d}\,\boldsymbol{w}_t$, and $\mathrm{d}\,\boldsymbol{h}_{t+1}/\mathrm{d}\,\boldsymbol{h}_t$ are given in (79), (78), and (80), respectively; and the blocks in the first row depend on the meta update.

## B. Exiting Step-size Optimization Algorithms as Special Cases of MetaOptimize

In this appendix we show that some of the existing step-size optimization algorithms are special cases of the MetaOptimize framework. In particular, we first consider the IDBD algorithm (Sutton, 1992) and its extension (Xu et al., 2018), and then discuss about the HyperGradient algorithm (Baydin et al., 2017).

## B.1. IDBD and Its Extensions

(Sutton, 1992) proposed the IDBD algorithm for step-size optimization of a class of quadratic loss functions. In particular, it considers loss functions of the form

$$f_t(\boldsymbol{w}_t) = \frac{1}{2}\left(\boldsymbol{a}_t^T\boldsymbol{w}_t - b_t\right)^2, \tag{82}$$

for a given sequence of feature vectors $\boldsymbol{a}_t$ and target values $b_t$, for $t = 1, 2, \ldots$. Moreover, Sutton (1992) assumes weight-wise step sizes, in which case $\boldsymbol{\beta}_t$ has the same dimension as $\boldsymbol{w}_t$. The update rule of IDBD is as follows:

$$\boldsymbol{g}_t \leftarrow \left(\boldsymbol{a}_t^T\boldsymbol{w}_t - b_t\right)\boldsymbol{a}_t, \tag{83}$$

$$\boldsymbol{\beta}_{t+1} \leftarrow \boldsymbol{\beta}_t - \eta\,\boldsymbol{h}_t\,\boldsymbol{g}_t, \tag{84}$$

$$\boldsymbol{\alpha}_{t+1} \leftarrow \exp\left(\boldsymbol{\beta}_{t+1}\right), \tag{85}$$

$$\boldsymbol{w}_{t+1} \leftarrow \boldsymbol{w}_t - \boldsymbol{\alpha}_{t+1}\,\boldsymbol{g}_t, \tag{86}$$

$$\boldsymbol{h}_{t+1} \leftarrow \left(1 - \boldsymbol{\alpha}_{t+1}\boldsymbol{a}_t^2\right)^+ \boldsymbol{h}_t - \boldsymbol{\alpha}_{t+1}\boldsymbol{g}_t, \tag{87}$$

where $(\cdot)^+$ clips the entries at zero to make them non-negative, aimed to improve stability. Here, $\boldsymbol{g}_t$ is the gradient of $f_t(\boldsymbol{w}_t)$ and $\boldsymbol{a}_t^2$ in the last line is a vector that contains diagonal entries of the Hessian of $f_t$. The updated values of $\boldsymbol{\beta}$ and $\boldsymbol{w}$ would remain unchanged, if instead of the vector $\boldsymbol{h}_t$, we use a diagonal matrix $\mathcal{H}_t$ and replace (84) and (87) by

$$\boldsymbol{\beta}_{t+1} \leftarrow \boldsymbol{\beta}_t - \eta\,\mathcal{H}_t\,\boldsymbol{g}_t,$$
$$\mathcal{H}_{t+1} \leftarrow \left(1 - \left[\boldsymbol{\alpha}_{t+1}\boldsymbol{a}^2\right]\right)^+ \mathcal{H}_t - \left[\boldsymbol{\alpha}_{t+1}\boldsymbol{g}_t\right]. \tag{88}$$

Note that $\left[\boldsymbol{a}^2\right]$ is a matrix that is obtained from zeroing-out all non-diagonal entries of the Hessian matrix of $f_t$. It is easy to see that the above formulation of IDBD, equals the L-approximation of MetaOptimize framework when we use SGD for both base and meta updates, and further use a diagonal approximation of the Hessian matrix along with a rectifier in the update of $\mathcal{H}_t$.

An extension of IDBD beyond quadratic case has been derived in (Xu et al., 2018). Similar to IDBD, they also consider weight-wise step sizes, i.e., $m = n$. The update of step sizes in this method is as follows:

$$\boldsymbol{\beta}_{t+1} \leftarrow \boldsymbol{\beta}_t - \eta\,\mathcal{H}_t^{\mathsf{T}}\,\nabla f_t(\boldsymbol{w}_t)$$
$$\boldsymbol{\alpha}_{t+1} \leftarrow \exp(\boldsymbol{\beta}_{t+1}),$$
$$\boldsymbol{w}_{t+1} \leftarrow \boldsymbol{w}_t - \boldsymbol{\alpha}_{t+1}\,\nabla f_t(\boldsymbol{w}_t),$$
$$\mathcal{H}_{t+1} \leftarrow \left(I - \left[\boldsymbol{\alpha}_{t+1}\right]\nabla^2 f_t(\boldsymbol{w}_t)\right)\mathcal{H}_t - \left[\boldsymbol{\alpha}_{t+1}\nabla f_t(\boldsymbol{w}_t)\right].$$

Similar to IDBD, it is straightforward to check that the above set of updates is equivalent to the L-approximation of MetaOptimize framework that uses SGD for both base and meta updates, except for the fact that the above algorithm uses $\boldsymbol{\alpha}_{t+1}$ in $\boldsymbol{w}_{t+1}$ and $\mathcal{H}_{t+1}$ updates whereas MetaOptimize uses $\boldsymbol{\alpha}_t$. This however has no considerable impact since $\boldsymbol{\alpha}_t$ varies slowly.

## B.2. Hyper-gradient Descent

HyperGradient descent was proposed in (Baydin et al., 2017) as a step-size optimization method. It considers scalar step size with straightforward extensions to weight-wise step sizes, and at each time $t$, updates the step size in a direction to minimize the immediate next loss function. In particular, they propose the following additive update for step sizes, that can wrap around an arbitrary base update:

$$\boldsymbol{\alpha}_t = \beta_t\,\mathbf{1}_{n\times 1},$$
$$\beta_{t+1} = \beta_t - \eta\,\frac{\mathrm{d}\,f_t(\boldsymbol{w}_t)}{\mathrm{d}\,\beta_{t-1}} = \beta_t - \eta\,\nabla f_t(\boldsymbol{w}_t)^T\,\frac{\mathrm{d}\,\boldsymbol{w}_t}{\mathrm{d}\,\beta_{t-1}}. \tag{89}$$

The last update can be equivalently written as

$$\beta_{t+1} = \beta_t - \eta\,\mathcal{H}_t^T\,\nabla f_t(\boldsymbol{w}_t),$$
$$\mathcal{H}_{t+1} = 0 \times \mathcal{H}_t + \frac{\mathrm{d}\,\boldsymbol{w}_{t+1}}{\mathrm{d}\,\beta_t}. \tag{90}$$

The step-size update in (90) can be perceived as a special case of MetaOptimize in two different ways. First, as a MetaOptimize algorithm that uses SGD as its meta update and approximate the $G_t$ matrix in (9) by zeroing out all of its blocks except for the top two blocks in the first column. From another perspective, the additive HyperGradient descent in (90) is also equivalent to a MetaOptimize algorithm that uses SGD as its meta update and sets $\gamma = 0$. Note that setting $\gamma$ equal to zero would eliminate the dependence of $\mathcal{H}_{t+1}$ on $X_t$ and $Q_t$, as can be verified from (14). This would also render the $\beta$ updates ignorant about the long-term impact of step size on future losses.

## C. Experiment Details

In the appendix, we describe the details of experiments performed throughout the paper. In our experiments on CIFAR10, non-stationary CIFAR100, and ImageNet dataset, we used a machine with four Intel Xeon Gold 5120 Skylake @ 2.2GHz CPUs and a single NVIDIA V100 Volta (16GB HBM2 memory) GPU. For TinyStories dataset, we used a machine with four AMD Milan 7413 @ 2.65 GHz 128M cache L3 CPUs and a single NVIDIA A100SXM4 (40 GB memory) GPU. In all experiments, the meta step size $\eta$ is set to $10^{-3}$. The meta-parameters used in the considered optimization algorithm for CIFAR10, non-stationary CIFAR100, ImageNet, and TinyStories are given in Table 2, Table 4, and Table 5, respectively. In the experiments, we performed a grid search for $\rho, \bar{\rho} \in \{0.9, 0.99, 0.999\}$, $\lambda, \bar{\lambda} \in \{0.99, 0.999\}$, and $c, \bar{c} \in \{0.9, 0.99\}$. Regarding baselines with fixed step sizes, we did a grid search for the learning rate in the set $\{10^{-5}, 10^{-4}, 10^{-3}, 10^{-2}, 10^{-1}\}$. We set $\gamma$ equal to one in all experiments. Moreover, in ImageNet (respectively TinyStories) dataset, for AdamW with the learning rate scheduler, we considered a cosine decay with 10k (respectively 1k) steps warmup (according to extensive experimental studies in (Chen et al., 2023) (respectively (Karpathy, 2024))) and did a grid search for the maximum learning rate in the set $\{10^{-5}, 10^{-4}, 10^{-3}\}$.

Regarding other baseline algorithms, for DoG, although it is a parameter-free algorithm, its performance is still sensitive to the initial step movement. We did a grid search for the initial step movement in the set $\{10^{-9}, 10^{-8}, 10^{-7}, 10^{-6}\}$ and reported the performance for the best value. In all experiments of DoG, we considered the polynomial decay averaging. For Prodigy, we used the default values of parameters as suggested by the authors in github repository. For gdtuo, we considered the following (base, meta) combinations: (RMSprop, Adam), (Adam, Adam), and (SGD with momentum, Adam) and chose the best combination. For mechanic, we did experiments for the base updates of SGDm, Lion, and Adam and considered the best update. In order to have a fair comparison, in mechanic and gdtuo, we used the same initial step size as MetaOptimize.

Regarding the complexity overheads reported in Table 1, for AdamW with fixed step-size we used the Pytroch implementation of AdamW. For all other baselines, we used the implementation from the Github repository provided along with (and cited in) the corresponding paper. For MetaOptimize, we used the implementation in (Anonymous, 2024). Note that the implementation of MetaOptimize in (Anonymous, 2024) is not optimized for time or space efficiency, and smaller complexity overheads might be achieved with more efficient codes. For each algorithm, the wall-clock time overhead and GPU space overhead are computed by $(T_{\text{Alg}} - T_{\text{AdamW}})/T_{\text{AdamW}}$ and $(B_{\text{AdamW}}^{\max}/B_{\text{Alg}}^{\max}) - 1$, respectively; where $T_{\text{Alg}}$ and $T_{\text{AdamW}}$ are per-iteration runtimes of the algorithm and AdamW, and $B_{\text{Alg}}^{\max}$ and $B_{\text{AdamW}}^{\max}$ are the maximum batch-sizes that did not cause GPU-memory outage for the algorithm and AdamW.

| Base Update | Meta Update (if any) | $\rho$ | $\lambda$ | $\kappa$ | $c$ | $\bar{\rho}$ | $\bar{\lambda}$ | $\bar{c}$ | $\alpha_0$ | $\eta$ | $\gamma$ |
|---|---|---|---|---|---|---|---|---|---|---|---|
| | Fixed step size | 0.9 | 0.999 | 0.1 | - | - | - | - | $10^{-5}$ | - | 1 |
| AdamW | Adam, Scalar | 0.9 | 0.999 | 0.1 | - | 0.9 | 0.999 | - | $10^{-6}$ | $10^{-3}$ | 1 |
| | Adam, Blockwise | 0.9 | 0.999 | 0.1 | - | 0.9 | 0.999 | - | $10^{-6}$ | $10^{-3}$ | 1 |
| | Fixed step size | 0.99 | - | 0.1 | 0.9 | - | - | - | $10^{-4}$ | - | 1 |
| Lion | Lion, Scalar | 0.99 | - | 0.1 | 0.9 | 0.99 | - | 0.9 | $10^{-6}$ | $10^{-3}$ | 1 |
| | Lion, Blockwise | 0.99 | - | 0.1 | 0.9 | 0.99 | - | 0.9 | $10^{-6}$ | $10^{-3}$ | 1 |
| | Fixed step size | - | 0.999 | 0.1 | - | - | - | - | $10^{-5}$ | - | 1 |
| RMSprop | Adam, Scalar | - | 0.999 | 0.1 | - | 0.9 | 0.999 | - | $10^{-6}$ | $10^{-3}$ | 1 |
| | Adam, Blockwise | - | 0.999 | 0.1 | - | 0.9 | 0.999 | - | $10^{-6}$ | $10^{-3}$ | 1 |
| | Fixed step size | 0.9 | - | 0.1 | - | - | - | - | $10^{-3}$ | - | 1 |
| SGDm | Adam, Scalar | 0.9 | - | 0.1 | - | - | - | - | $10^{-6}$ | $10^{-3}$ | 1 |
| | Adam, Blockwise | 0.9 | - | 0.1 | - | - | - | - | $10^{-6}$ | $10^{-3}$ | 1 |

*Table 2.* The values of meta-parameters used in CIFAR10 dataset.

| Base Update | Meta Update (if any) | $\rho$ | $\lambda$ | $\kappa$ | $\bar{\rho}$ | $\bar{\lambda}$ | $\alpha_0$ | $\eta$ | $\gamma$ |
|---|---|---|---|---|---|---|---|---|---|
| AdamW | Fixed step size | 0.9 | 0.999 | 0.1 | - | - | $10^{-4}$ | - | - |
| | Adam, Scalar | 0.9 | 0.999 | 0.1 | 0.9 | 0.999 | $10^{-4}$ | $10^{-3}$ | 0.999 |
| | Adam, Blockwise | 0.9 | 0.999 | 0.1 | 0.9 | 0.999 | $10^{-4}$ | $10^{-3}$ | 0.999 |

*Table 3.* The values of meta-parameters used in non-stationary CIFAR100 experiment.

| Base Update | Meta Update | $\rho$ | $\lambda$ | $\kappa$ | $c$ | $\bar{\rho}$ | $\bar{\lambda}$ | $\bar{c}$ | $\alpha_0$ | $\eta$ | $\gamma$ |
|---|---|---|---|---|---|---|---|---|---|---|---|
| AdamW | Fixed step size | 0.9 | 0.999 | 0.1 | - | - | - | - | $10^{-5}$ | - | 1 |
| | Lion, Scalar | 0.9 | 0.999 | 0.1 | - | 0.99 | - | 0.9 | $10^{-6}$ | $10^{-3}$ | 1 |
| | Lion, Blockwise | 0.9 | 0.999 | 0.1 | - | 0.99 | - | 0.9 | $10^{-6}$ | $10^{-3}$ | 1 |
| Lion | Fixed step size | 0.99 | - | 0.1 | 0.9 | - | - | - | $10^{-5}$ | - | 1 |
| | Lion, Scalar | 0.99 | - | 0.1 | 0.9 | 0.99 | - | 0.9 | $10^{-6}$ | $10^{-3}$ | 1 |
| | Lion, Blockwise | 0.99 | - | 0.1 | 0.9 | 0.99 | - | 0.9 | $10^{-6}$ | $10^{-3}$ | 1 |
| SGDm | Lion, Scalar | 0.9 | - | 0.1 | 0.9 | - | - | - | $10^{-5}$ | $10^{-3}$ | 1 |

*Table 4.* The values of meta-parameters used in ImageNet dataset.

# D. Further Experimental Results

**Continual CIFAR100 experiment:** In Figure 3, we plotted the average top-1 accuracy—averaged over all past training times. This metric is used in continual learning mainly because it summarizes algorithmic performance across multiple tasks, avoiding difficult/misleading interpretations from task-specific accuracy variations. Here, we also include the accuracy curves to reveal such variations. Figure 7 depicts test accuracy at the end of each task.

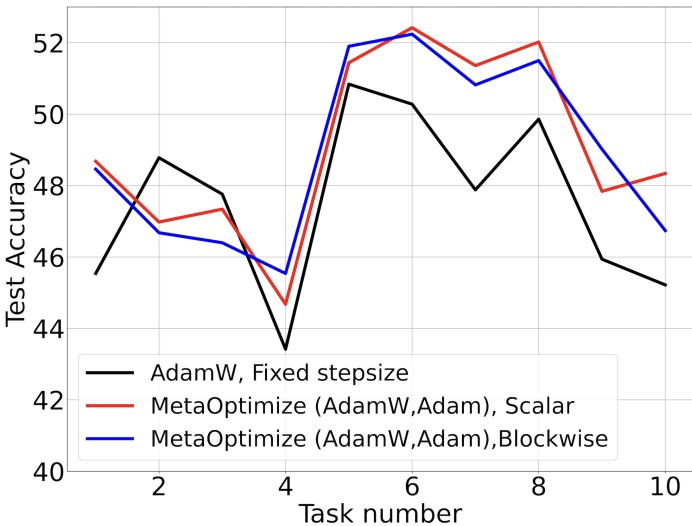

*Figure 7.* Top 1 test accuracy of each task in the non-stationary CIFAR100 experiment, computed at the end of the task.
.

**ImageNet dataset:** In Figure 8, we depict the train accuracy (top 1) and test accuracy (top 1) of the considered algorithms in ImageNet dataset. As can be seen, in the train accuracy (top 1), MetaOptimize (SGDm, Lion) and MetaOptimize (AdamW, Lion) have the best performance. Moreover, in the test accuracy (top1), these two combinations of MetaOptimze outperform other hyperparameter optimization methods and only AdamW with a handcrafted learning rate scheduler has a slightly better performance at the end of the training process.

Figure 9 shows the results for the blockwise version of MetaOptimize for two combinations of (AdamW, Lion) and (Lion, Lion). As can be seen, they showed no improvement over the scalar version.

| Base Update | Meta Update (if any) | $\rho$ | $\lambda$ | $\kappa$ | $c$ | $\bar{\rho}$ | $\bar{\lambda}$ | $\bar{c}$ | $\alpha_0$ | $\eta$ | $\gamma$ |
|---|---|---|---|---|---|---|---|---|---|---|---|
| AdamW | Fixed stepsize | 0.9 | 0.999 | 0.1 | - | - | - | - | $10^{-5}$ | - | 1 |
| | Adam, Scalar | 0.9 | 0.999 | 0.1 | - | 0.9 | 0.999 | - | $10^{-6}$ | $10^{-3}$ | 1 |
| Lion | Fixed stepsize | 0.99 | - | 0.1 | 0.9 | - | - | - | $10^{-4}$ | - | 1 |
| | Lion, Scalar | 0.99 | - | 0.1 | 0.9 | 0.99 | - | 0.9 | $10^{-6}$ | $10^{-3}$ | 1 |

*Table 5.* The values of meta-parameters used in TinyStories dataset.

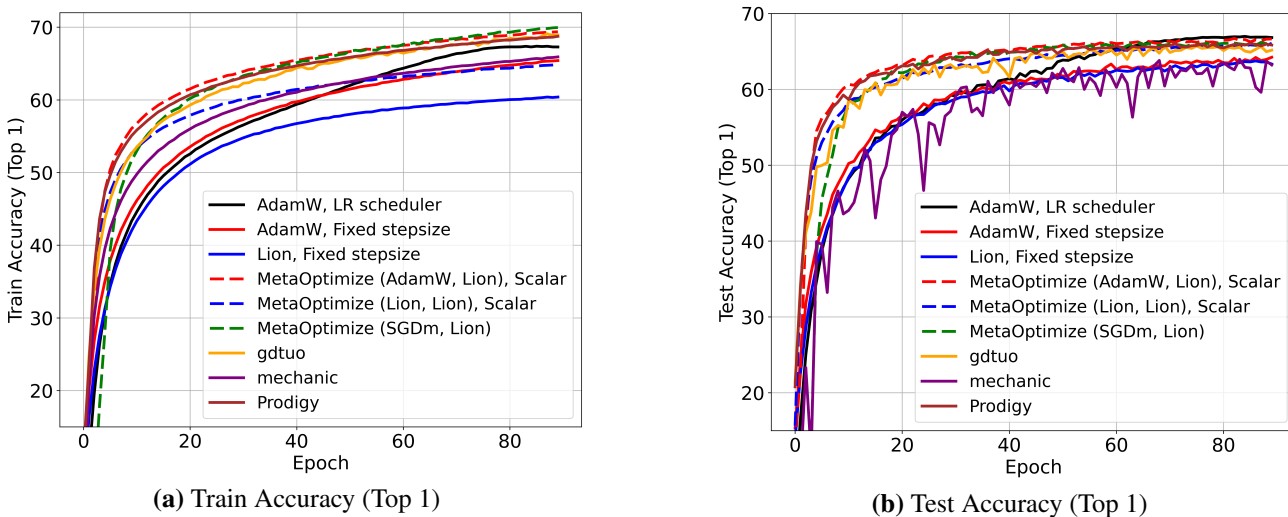

**(a)** Train Accuracy (Top 1)

**(b)** Test Accuracy (Top 1)

*Figure 8.* ImageNet learning curves.

**TinyStories experiment:** In Figure 10, we provide the test loss of considered algorithms for the TinyStories datasets. As can be seen, the learning curves have the same trends as the training loss in Figure 6.

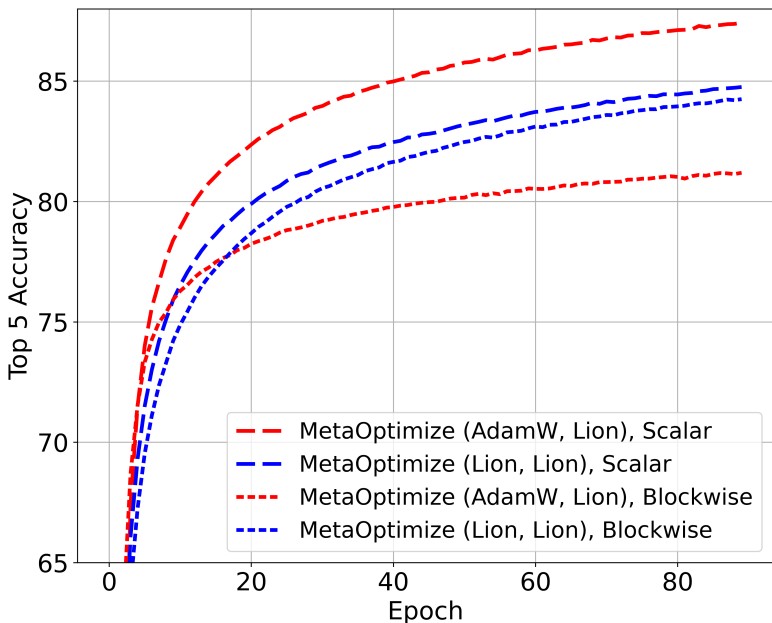

*Figure 9.* Comparison of blockwise version of MetaOptimize with the scalar version in ImageNet dataset.

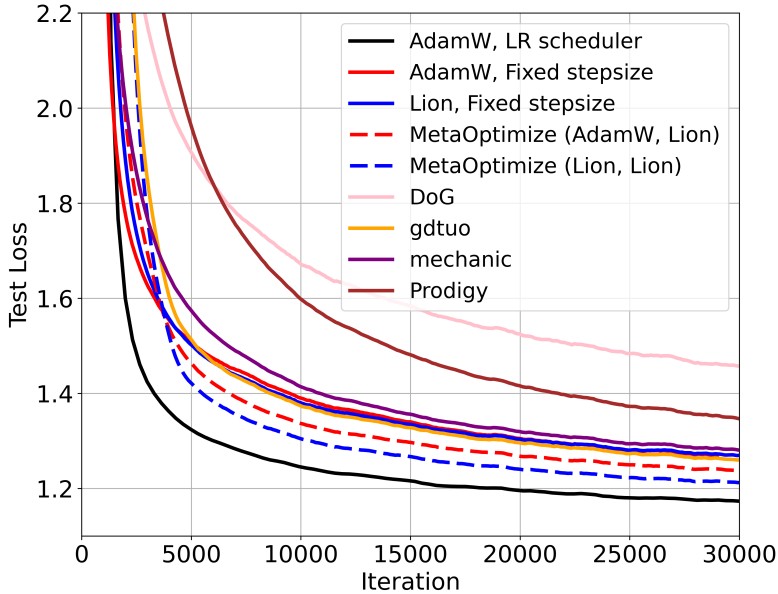

*Figure 10.* TinyStories learning curves.

