# OpenReview forum: "MetaOptimize: A Framework for Optimizing Step Sizes and Other Meta-parameters"
_ICML.cc/2025/Conference — ICML 2025 poster_

### Official Review · Reviewer_TH46 · 2025-02-27

**Overall Recommendation:** 1

**Summary:**

This paper proposes to optimize step size by viewing step size as a differentiable parameter, and minimizing the objective of cumulated loss by recording a temporal trajectory.

**Claims And Evidence:**

Yes

**Essential References Not Discussed:**

No

**Experimental Designs Or Analyses:**

No.

**Methods And Evaluation Criteria:**

No. Using eqn (5) as a surrogate of eqn (4) is the basic to make the proposed method practical. However, such approximation is not reasonable. I can not see any rationality in footnote 1 as justification of such approximation. If as it says, $\beta_T^5=\beta_T^4$, then $\beta_T^5=\beta_T^4=\beta_0$: we can not update $\beta$ under such assumption.

**Other Comments Or Suggestions:**

I am not familiar with related works in this area at all, and I feel confused why OpenReview assign this paper to me to review.

But I can understand the proposed method and find fatal error as mentioned above.

**Other Strengths And Weaknesses:**

None

**Questions For Authors:**

None.

**Relation To Broader Scientific Literature:**

I could not identify the contribution as the proposed method can not be understood.

**Theoretical Claims:**

Yes. Please refer to Methods And Evaluation Criteria

---

> ### Author Rebuttal · Authors · 2025-03-31
>
> We appreciate the reviewer's honesty in stating clearly that this paper falls outside their area of expertise. However, it seems there was a critical misunderstanding regarding the approximation in Equation (5).
>
> > **Reviewer:** _"Using eqn (5) as a surrogate of eqn (4) is the basic to make the proposed method practical. However, such approximation is not reasonable. I cannot see any rationality in footnote 1 as justification..."_
>
> We clarify the approximation carefully below, as there appears to be a misunderstanding:
>
> - The statement in Footnote 1 (p.2) specifically analyzes the case as $\eta \to 0$. In this scenario, indeed $\beta_T^5 - \beta_T^4 \to 0$. However, the non-trivial insight is that $(\beta_T^5 - \beta_T^4)/\eta \to 0$, meaning the difference between these terms vanishes faster than $\eta$. Concretely, this implies:
>   - While both $\beta_T^4$ and $\beta_T^5$ scale roughly as $\eta T$, their difference scales strictly slower ($o(\eta T)$), making the approximation increasingly accurate for small $\eta$.
>
> - Intuitively, each term in the RHS of Eq. (5) directly appears in Eq. (4), though at the earliest feasible time it can be computed (ensuring causality). Thus, Eq. (5) serves as a valid practical approximation to Eq. (4).
>
> Finally, we emphasize that this forward-backward view approximation is well-established in related fields, notably in eligibility trace methods widely used in reinforcement learning literature.

---

### Official Review · Reviewer_51Lj · 2025-03-12

**Overall Recommendation:** 4

**Summary:**

This paper proposes a method for optimising hyperparameters online in first order optimisation algorithms. Using this framework, performance is drastically improved over using fixed hyperparameters in a number of experiments. There are a number of qualitative approximations used which can simplify the complexity of the method.

**Claims And Evidence:**

The papers claims are as follows:
- They produce a formal approach for optimising step-size and other hyperparameters. This is true.
- Their algorithm is general and can be applied to any first-order optimisation algorithm. This is true.
- Their algorithm is computationally efficient thanks to a number of approximations. It is true that there are a number of approximations, and a small demonstration that their method does not have too large an overhead; it would be interesting to see compute-normalised experiments too, although this may not be practical.
- They demonstrate that some prior work is purely specific instances of this more general framework, which is true. I would, however, be curious to see how these specific instances compare to the algorithm used in this work empirically.

**Essential References Not Discussed:**

See above.

**Experimental Designs Or Analyses:**

All experiments seem to be run for a single seed, without error bars. Besides this, the apper includes hyperparameters and meta-parameters needed for the experiments, and seems like it should be fully reproducible. There is a relatively brief sensitivity analysis, and a larger analysis considering how the step size of a model optimised using MetaOptimise changes over training. I believe that the necessary analyses are covered; more is always better, but I think the necessary bases are covered.

**Methods And Evaluation Criteria:**

There are 3 algorithms, growing in specificity, proposed in this work. These are sensible applications from the more theoretical framework generated earlier in the paper. They compare against a number of sensible baselines - in many cases, using a standard learning rate, but also occasionally against other online hyperparameter optimisation algorithms. The one comparison I think is missing is between their implementation and the specific instances of their framework (i.e. hypergradient descent and IDBD, which are discussed in the paper). I would be curious to know whether their more general system enables improved performance.

Their method is evaluated on a number of standard stationary datasets, which is good. However, I think the missing benchmark that would really complete this paper would be in the method's application to continual learning problems, where having adaptive hyperparameters could lead to huge performance improvements but which may prove difficult to operate in; the paper discusses that MetaOptimise can be applied to continual learnign problems but as far as I can tell it is not evaluated on any.

**Other Comments Or Suggestions:**

I think most of my other comments are interspersed into the review.

**Other Strengths And Weaknesses:**

Strengths:
- I liked the plots showing robustness to idfferent meta-parameters (i.e. the initial alpha). I also think it is interesting to see that the step size differs between the first and second block.

Weaknesses:
- In figure 3, average cumulative accuracy is a very strange metric. Why not report the accuracy?
- I would love to see more discussion around figure 3 and 4 - in particular, why you think this happens. To properly draw conclusiosn, I would want to see this phenomena occur at least over multiple seeds but, preferably, over multiple different experiments (eg the second block step size growing through training). I also think that the way results in figure 4 are displayed is a bit misleading, since the two lines use different scales - I think a naive reader might think the step sizes are crossing over, but it is actually more interesting to note that the second block has much larger step sizes!

**Questions For Authors:**

- Why do you think the second block needs larger learning rates, and grows over time?
- How stable is your method?
- How would your method work in a non-stationary dataset that is standard for continual learning?

**Relation To Broader Scientific Literature:**

The paper contextualises prior literature as specific instances of its more general framework. This is done rigorously. There is a large related work section which seems to provide good coverage of prior literature, albeit missing one crucial field in my opinion.

I personally think contextualisation of this work against learned optimisation would provide additional value - in L2O, the optimiser is learned upfront and then not changed at test-time, which provides an interesting comparison to this work where the hyperparameters are learned at the same time as the model.

I highlight a few referencess below with some reasons why I think they should specifically be included:

- Learning to learn by gradient descent by gradient descent, 2016, Andrychowicz et al. - kicked off a lot of the L2O work.
- Tasks, stability, architecture, and compute: Training more effective learned optimizers, and using them to train themselves, 2020, Metz et al. - a strong demonstration of learned optimisers
- VeLO: Training Versatile Learned Optimizers by Scaling Up, 2022, Metz et al. - a large scale learned optimiser with no hyperparameters
- Practical tradeoffs between memory, compute, and performance in learned optimizers, 2022, Metz et al. - Introduces a python library for learned optimisation which, among others, includes an optimiser 'NN_Adam' where the hyperparameters of Adam are replaced with a neural network.
- Can Learned Optimization Make Reinforcement Learning Less Difficult, 2024, Goldie et al. - a learned optimiser designed for reinforcement learning which exhibits much of the desired behaviour in MetaOptimise, such as having different update sizes for different parts of the network and a step size which changes throughout the course of training, including at the start of every new batch (which I assume is what causes the periodic structure in figure 3 and 4).

**Theoretical Claims:**

There are a number of derivations, which I followed through in the main body of the paper, but there are not theoretical proofs.

---

> ### Author Rebuttal · Authors · 2025-03-31
>
> We thank the reviewer for their thoughtful feedback and helpful suggestions, which we address below and plan to incorporate into the final version of the paper. We believe this will significantly enhance the clarity and quality of the work.
>
> ---
>
> ### Continual Learning
> >I think the missing benchmark that would really complete this paper would be in the method's application to continual learning problems... the paper discusses that MetaOptimize can be applied to continual learning problems but... it is not evaluated on any.
>
> We already include a continual learning experiment in Section 7.2 on continual CIFAR-100 benchmark. This benchmark sequentially trains on 10 tasks (10 classes each) without explicit task boundaries or weight resets. As discussed in the paragraph preceeding Figures 3 and 4, MetaOptimize demonstrates substantial advantages in continual learning, including fast adaptation to new tasks, and favourable layer-wise adaptive behaviour.
>
> >I would love to see more discussion around figure 3 and 4—in particular, why you think this happens.
>
> Here are some intuition for the observed phenomena:
>
> - **Why later-layer step-sizes are consistently larger:**
> This aligns with established best practices in supervised learning (Howard & Ruder, 2018), where larger step-sizes in later layers accelerate training. MetaOptimize autonomously discovers this pattern, despite equal initial step-sizes (10⁻⁴).
>
> - **Why step-sizes of earlier layers decrease and later-layers increase in continual learning:**
> Early layers converge toward stable low-level image features shared across tasks, thus needing smaller updates over time. Later layers must continually adapt to changing labels, thus requiring constantly larger step-sizes.
>
> - **Saw-tooth pattern in step-sizes (Fig. 4):**
> Each spike corresponds exactly to task transitions. MetaOptimize momentarily boosts step-sizes to facilitate rapid adaptation to new tasks.
>
> >I would want to see this phenomenon occur... over multiple seeds...
>
> Figures 3 and 4 report averages over **5 random seeds**, an important detail we mistakenly omitted in the submitted manuscript. We will explicitly clarify this in our revision.
>
> ---
>
> ### Methods and Evaluations
>
> >The one comparison I think is missing is between their implementation and the specific instances of their framework (i.e. hypergradient descent and IDBD).
>
> Thank you for pointing this out. Hypergradient Descent is already included in our experiments as gdtau (Chandra et al., 2022), an efficient implementation of hypergradient descent.
>
> Regarding IDBD, its original design targets linear regression tasks, making it inapplicable to the neural-network-based scenarios presented in our work.
>
> >In figure 3, average cumulative accuracy is a very strange metric. Why not report the accuracy?
>
> Average cumulative accuracy is used in continual learning mainly because it summarizes algorithmic performance across multiple tasks, avoiding difficult/misleading interpretations from task-specific accuracy variations. In the final version of the paper, we will also include the accuracy curves to reveal such variations.
>
> >figure 4 is a bit misleading, since the two lines use different scales.
>
> Thank you for highlighting this. Initially, both blocks have identical step-sizes (10⁻⁴). To eliminate confusion, we will plot step-sizes for the two blocks in separate subfigures.
>
> ---
>
> ### Relation to Broader Scientific Literature (Learned Optimization)
>
> Thank you for highlighting the interesting relevant works on Learned-to-Optimize (L2O). Traditional L2O approaches typically learn optimizers or hyperparameters offline, then fix them at deployment. In contrast, MetaOptimize dynamically adjusts hyperparameters (step sizes) concurrently during training. This real-time adaptation to changing conditions or tasks is specifically beneficial in continual learning. In the revised manuscript, we will carefully contrast MetaOptimize with L2O, explicitly citing and discussing the references you kindly suggested. Additionally, we are thinking that a combination of the two areas (e.g., learning to meta optimize, or using L2O-discovered optimizer as base algorithms MetaOptimize, etc) might be promising research directions.
>
> The reference on L2O for RL is particularly interesting. While the present manuscript focuses on supervised and continual learning (as more controlled settings), we plan future research extending MetaOptimize explicitly into RL.
>
> ---
>
> ### Stability of MetaOptimize
>
> >How stable is your method?
>
> All reported MetaOptimize experiments exhibit stable, consistent behavior across multiple random seeds. However, we observed instability in certain variants not included in the experiments (e.g., weight-wise or layer-wise step sizes in very deep networks). We explicitly mention these stability challenges in our "Limitations" section, suggesting directions for future research. Solving these underlying issues could significantly enhance MetaOptimize performance.

---

> > ### Comment · Reviewer_51Lj · 2025-04-02
> >
> > Dear Authors,
> >
> > Thank you for your response.
> >
> > Re: continual learning problems - I apologise for my oversight, you are right that this is clearly evaluated on a non-stationary learning problem. I think when I was writing my review I had something specific in mind, but I can't figure out what that could ahve been now as that concern is covered by 7.2. Apologies.
> >
> > Re: intuition, this is very interesting - it would be good to discuss this in the paper if there is time and space.
> >
> > Re: gdtuo, perhaps it would be good to make this explicit in the work, particularly for someone (like me) who is unfamiliar with this direct line of research
> >
> > Re: Evaluation metrics, I think including the current accuracy would be good to demonstrate the performance over time. I trust that these will be included in the end product, though I am obviously unable to judge what the curves look like since ICML does not allow sharing any additional drafts.
> >
> > Re: fig 4, I didn't actually realise both blocks had identical step sizes. I do like the inclusion of both lines on the same plot, but think that two separate subfigures might make the story clear - perhaps it could be good to put the current figure 4 in the appendix somehow as well?
> >
> > Re: Literature, Feel free to take my suggestion or not on these references - it is the world that I come from so I found it a natural fit, but that may not be the case for you and I believe you have adequately contextualised your work in its relevant field - I have not factored that into my score, I just thought it might provide some nice contrast.
> >
> > Overall, I think that a number of my concerns will have been rectified in the camera-ready version of hte paper, but I am unable to currently factor that into my review given I will not have seen these changes. Overall, I think this is a good, high quality paper that poses an interesting research question. I do not belief it has the 'wow' factor that would take it to a 5, which I think is reserved for truly groundbreaking papers, but I do believe that this paper is deserving of acceptance to ICML and thus uphold my score of 4.

---

> > > ### Author Response · Authors · 2025-04-07
> > >
> > > We sincerely thank you for taking the time to reflect and share your perspective — your feedback has been very helpful. We will ensure that all the promised modifications will be incorporated into the camera-ready version.
> > > Given the potential of meta optimization, we hope this work can serve as a stepping stone toward future research that achieves truly groundbreaking results and delivers the "wow" factor.

---

### Official Review · Reviewer_wz4K · 2025-03-17

**Overall Recommendation:** 2

**Summary:**

This paper introduces MetaOptimize, an approach for automatically adapting the learning rates of base optimization algorithms like SGD, Adam, and Lion.  MetaOptimize maintains a set of learning rates updated with a separate stochastic gradient optimizer to minimize the discounted sum of future losses.  Naively updating the learning rates requires maintaining multiple states and computing the Hessian of the base objective; the authors propose a tractable Hessian-free formulation with multiple approximations to reduce computational overhead.  Experiments show MetaOptimize outperforms previous meta-optimization methods like Prodigy and mechanic and matches a well tuned AdamW for ResNet on Imagenet and outperforms AdamW with a fixed learning rate for GPT on TinyStories (lags well-tuned AdamW).

**Claims And Evidence:**

Figure 2 and 3 provide good evidence that MetaOptimize is able to recover from poor initial learning rates and also adapt to nonstationary data.

Results in Figure 6 shows a well-tuned MetaOptimize outperforms prior approaches like Prodigy, mechanic, and gdtuo.  However, MetaOptimize is sensitive to the selection of the optimizer used for base and meta updates and still lags a well-tuned AdamW on TinyStories.

**Essential References Not Discussed:**

References to prior work is sufficiently discussed.

**Experimental Designs Or Analyses:**

The experimental design and analysis are sound.

**Methods And Evaluation Criteria:**

The evaluation methods make sense for the problem.

**Other Comments Or Suggestions:**

Given the suitability of MetaOptimize for continual learning and nonstationary data, I think the experiments would benefit from a practical nonstationary learning problem from RL or continual learning.

**Other Strengths And Weaknesses:**

Strengths:
- The paper is well written and easy to understand.

Weaknesses:
- In Section 7.5, the authors state that a discount factor of 1 was used in all stationary experiments and performance degrades meaningfully with $\gamma\leq 0.999$.  This makes the discounting of future losses somewhat contrived.
- It is unclear how valuable MetaOptimize is in practice given it lags a well-tuned AdamW on TinyStories by a meaningful amount.
- Experiments are on fairly small models with <20m parameters.  It is unclear how well MetaOptimize performs on larger models.
- The performance of MetaOptimize seems sensitive to the base and meta optimizers selected.  For example, base AdamW and Lion for MetaOptimize worked best on ImageNet but base Lion and Lion for MetaOptimize worked best on TinyStories.

**Questions For Authors:**

Does the backward formulation depend on the discount factor being less than 1?  How does a using a discount factor of 1 as is done for the stationary experiments impact the derivation?

**Relation To Broader Scientific Literature:**

This work follows prior work on learning-rate free optimizers and provides a formulation based on minimizing sum of discounted future losses.

**Theoretical Claims:**

There are no error bounds/convergence guarantees presented in the paper.  I did not check the derivation of MetaOptimze updates for different base optimizer.

---

> ### Author Rebuttal · Authors · 2025-03-31
>
> We thank Reviewer wz4K for thoughtful and valuable feedback. Below we provide clarifications, which we will also highlight in final version of the paper.
>
> ---
>
> ### On Discount Factor γ:
>
> >Authors state γ=1 was used in stationary experiments, and performance meaningfully degrades for γ<0.999, making discounting somewhat contrived.
>
> We clarify two important aspects:
>
> -  Our general framework naturally accommodates discount factors less than one. While stationary experiments favored γ≈1 due to implicit decay (discussed in the next bullet), having the flexibility to set γ<1 is beneficial for non-stationary tasks (e.g., continual learning scenarios, as shown in Section 7.2), where explicit discounting significantly improves performance. Thus, the framework’s generality allowing γ<1 is not contrived but genuinely beneficial for practical settings.
>
> -  The use of weight decay implicitly introduces discounting in meta-updates. To see why clearly, consider the base optimizer being SGD with weight decay parameter $\kappa$. The eligibility trace updates (under L approximation) become:
>   $$
>   h_{t+1} = \gamma (1 - \alpha_t \kappa) h_t - \gamma\alpha_t \nabla^2 f_t(w_t) h_t - \alpha_t \nabla f_t(w_t).
>   $$
>   Even if we set γ=1 and use a Hessian-free approximation (i.e., ignore the Hessian term), the trace $h_t$ still decays at rate $(1 - \alpha_t \kappa)$, exhibiting a discount-like effect. Thus, explicit discounting (γ<1) was unnecessary here primarily due to this implicit decay from weight decay.
>
> >Does backward formulation rely on γ<1? How does γ=1 impact derivation?
>
> This is an important subtlety, and you correctly identified that our backward derivation (Eq. (6)) formally assumes γ<1 because of the scaling term $(1-\gamma)$ used for normalization. A straightforward workaround for γ=1 is simply removing this scaling factor $(1-\gamma)$ from the definition and all subsequent formulas. This slight adjustment makes the derivation fully valid and consistent for all γ≤1, matching precisely what we implemented and tested experimentally.
>
> Your comment also highlights an interesting direction for future research. In RL and dynamic programming, the formulation and algorithms for the case γ=1 differ subtly, requiring additional  reward-centering. Exploring analogous techniques for meta-optimization when γ=1 could be beneficial and is a promising avenue we will include in our future work discussion.
>
> ---
>
> ### On Practical Value:
>
> >MetaOptimize lags behind well-tuned AdamW on TinyStories; thus, its practical value is unclear.
>
> We acknowledge this observation and provide some clarification:
>
>  - The AdamW baseline used a carefully tuned step-size schedule through extensive search, specifically optimized for this task; while MetaOptimize involved no manual tuning. Thus, we did not expect scalar MetaOptimize to outperform such an extensively optimized schedule.
>
> -  In addition to avoiding manual hyperparameter tuning, MetaOptimize benefits from generality and applicability to generic meta-paramters beyond stepsizes. The biggest advantage however appear in continual learning (see experiment in Section 7.2), where tasks evolve over time and pre-defined stepsize schedules are ineffective.
>
> - Crucially, we emphasize this paper’s primary contribution as foundational, laying the groundwork for a highly promising research direction. The greatest benefits of meta optimization methods are likely still ahead. We explicitly outline multiple clear pathways for future improvements (Section 9), highlighting that the current results are only the beginning. Thus, while initial performance is competitive rather than groundbreaking, the demonstrated potential and clear roadmap for future research represent the key strengths of this work.
>
> >Sensitivity to the base optimizer .., For example, base AdamW worked best on ImageNet while base Lion worked best on TinyStories.
>
> The key point is that regardless of the base optimizer, applying MetaOptimize to learn the step sizes consistently outperforms the same base algorithm with best fixed step size. Note that the  Lion optimizer tends to work better in training Transformers even when using fixed step sizes.
>
> ---
>
> ### Scale of Experiments:
> We agree scaling is important. Due to resource constraints, we focused on models <20M parameters, but included diverse architectures to show broad applicability. MetaOptimize is theoretically scale-agnostic and compatible with any first-order optimizer, so we expect it to extend naturally to larger models. Scaling up is a valuable next step and will be highlighted in future work.
>
> ---
>
> ### Continual Learning:
> Section 7.2 includes a continual CIFAR-100 experiment where MetaOptimize shows strong adaptation: step sizes increase after task switches to support quick adaptation, and decrease in early layers over time, reflecting stable feature reuse. These behaviors align well with continual learning needs, showcasing MetaOptimize’s potential in nonstationary settings.

---

### Official Review · Reviewer_Liw6 · 2025-03-19

**Overall Recommendation:** 4

**Summary:**

The proposed approach seeks to dynamically adjust hyper parameters during the optimization process. The MetaOptimize framework, which utilizes historical iteration data, seeks to minimize regret builds on a discounted sum of future losses. This framework, coupled with several approximations, calculates meta-gradients to update the hyper parameters. The approximations suggested in the paper significantly reduces computational overhead, particularly in relation to Hessian matrix calculations. To demonstrate its effectiveness, the new optimizer was evaluated on both visual and textual datasets. Performance metrics, including learning trajectories and computational efficiency, were analyzed and presented as evidence of the method's efficacy.

**Claims And Evidence:**

Yes

**Essential References Not Discussed:**

No

**Experimental Designs Or Analyses:**

Yes, the results look valid.

**Methods And Evaluation Criteria:**

Yes

**Other Comments Or Suggestions:**

On page 2, There are two font versions of  w_t in the text (see lines 73 and 84, for example) and I am not sure if they are the same.
On page 3, left column, line 121 , w_{t+1} --> x_{t+1}?
On page 3, left column, line 122, You don't need w_{t+1} for the computation of H_{T}^{T}\nabla f_{t}(w_{t}), right?
On page 3, right column, line 127, It seems you start a new sentence here as formula (13) ends with a period.
On page 10, last reference, RS Sutton --> Richard S Sutton :).
On page 12, line 630, Maybe you want to mention when you write \sigma'(\beta_{t})) the relation in (3) defining \alpha_t.
On page 13, line 682, inequality --> equality.
On page 16, line 833, there is a missing \gamma on the right hand side (unless I missed something).

**Other Strengths And Weaknesses:**

Strengths. This paper presents a novel and general MetaOptimize framework.
The framework is made practical through proposed approximation and Hessian-free versions that reduce computational complexity.
The effectiveness of the approach is thoroughly demonstrated through extensive experiments on both image classification and language modeling benchmarks

Weaknesses. The technique proposed in this paper is designed to handle the setting where we a sequence of loss functions {f_t}. However, this comes with the price of approximating the true gradient with a sum of the gradients over the past iterations. This idea follows the eligibility-trace-style approach, which is well-known in RL. Why the authors believe that this approach is also relevant in approximating gradients as it seems to be a very different task than RL? Can we measure the error produced by this approximation?

In reducing the complexity of the proposed optimizer, the authors suggest some approximations other matrix G_t. This is done on top of the approximation mentioned in the first point (there is also the approximation that is done in Section 5). This raise the question of comparing the technique proposed in this paper with other possible techniques for learning hyper parameters. For example, comparing to the recent paper MADA: Meta-Adaptive Optimizers through hyper-gradient Descent (ICML 2024) by Ozkara et. al..

In practical settings, stochastic optimizers are used to overcome the intractability of computing gradients. However, in this paper, the authors develop a deterministic meta optimizer. How a stochastic version would look like? In the numerical experiments, do you use the deterministic version that is presented in the paper?

The derivations in Section 4 are complicated to follow. I understand that space is always an issue but more explanations will be very helpful.

**Questions For Authors:**

See my list of weaknesses.

**Relation To Broader Scientific Literature:**

As far as I know the relevant literature, the idea proposed in this paper is novel.

**Theoretical Claims:**

There are no proofs in the paper, but there are some mathematical developments, which seems correct to me.

---

> ### Author Rebuttal · Authors · 2025-03-31
>
> We thank Reviewer Liw6 for their thoughtful and constructive comments. We carefully considered each suggestion and provide detailed clarifications below. We will apply the suggested improvements in the final version, which we believe significantly helps improve the clarity and presentation of the paper.
>
> ---
>
> ### Reviewer’s Concern 1: Relevance of Eligibility-Trace-Style Approximation
>
> > “The technique proposed...follows the eligibility-trace-style approach, well-known in RL. Why do the authors believe this RL-inspired approach is relevant to approximating gradients, as these seem like very different tasks?”
>
> The eligibility-trace-style approach is relevant and beneficial in our setting because it efficiently approximates the meta-gradient in a causal way: each term in the RHS of Eq. (5) directly appears in Eq. (4), though at the earliest feasible time it can be computed (ensuring causality). An alternative, would be direct gradient computation through finite differences which requires unrolling the optimization procedure far into the future at each iteration, resulting in prohibitive computational and numerical challenges; and also results in large delays in updates not acceptable in online and continual applications. In contrast, the eligibility-trace approach provides an instant and tractable approximation by summarizing gradient effects over time.
>
> As noted in the footnote on page 2, our approximation becomes exact when the meta-stepsize tends to zero. For larger meta-stepsizes, approximation accuracy naturally decreases—a phenomenon also observed in RL. In RL, sophisticated techniques like Dutch traces have been developed to mitigate such inaccuracies (van Hasselt et al., 2014). Similarly, extending these advanced RL-inspired methods to our meta-optimization framework is an exciting direction we highlight explicitly as future work (Section 9).
>
> ---
>
> ### Reviewer’s Concern 2: Comparison with Other Meta-learning Techniques (e.g., MADA)
>
> > “In reducing complexity, several approximations are used. It raises the question of comparing this technique explicitly with recent methods such as MADA (ICML 2024, Ozkara et al.).”
>
> The MADA method builds upon the Hypergradient-descent approach of Baydin et al. (2018), updating meta-parameters based on immediate hypergradients. We have already thoroughly compared MetaOptimize against Hypergradient-descent in our experiments (referred to as the "gdtau baseline" in Section 5 and Appendix B). Crucially, we demonstrate theoretically and empirically that Hypergradient-descent corresponds exactly to the special case of MetaOptimize with the discount parameter $\gamma=0$. Thus, MADA inherently shares this limitation by considering only immediate-step effects, whereas our proposed method explicitly models long-term impacts via nonzero $\gamma$.
>
> We appreciate your suggestion and will explicitly reference and clarify this comparison with MADA in the final version of the paper.
>
> ---
>
> ### Reviewer’s Concern 3: Deterministic vs. Stochastic Meta-Optimizer
>
> > “In practice, stochastic optimizers are employed due to gradient intractability, whereas your paper presents a deterministic meta-optimizer. How would a stochastic version look, and was the deterministic version used in experiments?”
>
> Thank you for highlighting this. We clarify that, despite the formal deterministic derivation in Section 4, our practical implementation is inherently stochastic. Specifically, the meta-update at each step (including the matrix $G_t$ computation) relies on gradients estimated using stochastic samples (such as mini-batches). Thus, in practice, MetaOptimize is already stochastic.
>
> If desired, additional stochasticity could be introduced by employing further random samples (e.g., from a validation set) within each meta-update. We avoided this approach in our experiments to minimize sample complexity.
>
> We will clarify the stochastic nature of our experimental implementation explicitly in the final paper revision.
>
> ---
>
> ### Reviewer’s Concern 4: Complexity of Section 4 Derivations
>
> > “Derivations in Section 4 are complex. More explanations would greatly improve readability.”
>
> Thank you for pointing out this readability issue. We will significantly enhance clarity in Section 4 by explicitly providing a concise outline of the derivations at the section's outset. Our goal is to clarify the key conceptual steps, making the mathematical reasoning easier to follow.
>
> ---
>
> ### Reviewer’s Minor Comments
>
> We are grateful for the reviewer’s meticulous attention to detail. We confirm that all minor suggestions (notation inconsistencies, typos, equation corrections) are valid and helpful. All indicated issues, including font inconsistencies, typographical errors, and minor mathematical clarifications, will be carefully fixed in the final version.

---

> > ### Comment · Reviewer_Liw6 · 2025-04-07
> >
> > Dear authors, thank you for your rebuttal. You have addressed my concerns. I understand your point regarding Eligibility-Trace-Style Approximation. I have also read through the other reviews and your responses. I think the paper should be accepted!

---

> > > ### Author Response · Authors · 2025-04-07
> > >
> > > Thank you for your valuable feedback. We will make sure that all the proposed modifications are thoroughly addressed in the camera-ready version.

---

### Decision · Program_Chairs · 2025-05-01

**Decision:**

Accept (poster)

**Comment:**

This paper proposes an online approach to hyper parameter optimisation. Unlike L2O approaches, hyper parameters are adjusted during training, which is particularly helpful in continual learning settings, for example. Reviewers generally agreed that this is an interesting and valuable line of work.
The outstanding divisive issue, which generated some debate during the discussion phase, is what scale of experiments should be required to empirically validate a new optimisation method. This paper considers ResNet18 (CIFAR, ImageNet) and a small language model. I agree that bigger experiments would be preferred and probably generate more attention for the work. However, it’s already interesting and hopefully other members of the community can help to apply it at scale.
I recommend accept.